# Nano-Particles Carried by Multiple Dynein Motors Self-Regulate Their Number of Actively Participating Motors

**DOI:** 10.3390/ijms22168893

**Published:** 2021-08-18

**Authors:** Gal Halbi, Itay Fayer, Dina Aranovich, Shachar Gat, Shay Bar, Vitaly Erukhimovitch, Rony Granek, Anne Bernheim-Groswasser

**Affiliations:** 1The Department of Chemical Engineering, Ben-Gurion University of the Negev, Beer Sheva 84105, Israel; galhalbi91@gmail.com (G.H.); dina.aranovich@gmail.com (D.A.); gats@post.bgu.ac.il (S.G.); shayba@post.bgu.ac.il (S.B.); evitaly@bgu.ac.il (V.E.); 2The Stella and Avram Goren-Goldstein Department of Biotechnology Engineering, Ben-Gurion University of the Negev, Beer Sheva 84105, Israel; itayfayer@gmail.com; 3The Ilse Katz Institute for Meso and Nanoscale Science and Technology, Ben-Gurion University of the Negev, Beer Sheva 84105, Israel

**Keywords:** active transport, multi-motor complex, nano-particles, motility assays, single particle tracking, Monte-Carlo simulations

## Abstract

Intra-cellular active transport by native cargos is ubiquitous. We investigate the motion of spherical nano-particles (NPs) grafted with flexible polymers that end with a nuclear localization signal peptide. This peptide allows the recruitment of several *mammalian* dynein motors from cytoplasmic extracts. To determine how motor–motor interactions influenced motility on the single microtubule level, we conducted bead-motility assays incorporating surface adsorbed microtubules and combined them with model simulations that were based on the properties of a single dynein. The experimental and simulation results revealed long time trajectories: when the number of NP-ligated motors *N_m_* increased, run-times and run-lengths were enhanced and mean velocities were somewhat decreased. Moreover, the dependence of the velocity on run-time followed a universal curve, regardless of the system composition. Model simulations also demonstrated left- and right-handed helical motion and revealed self-regulation of the number of microtubule-bound, actively transporting dynein motors. This number was stochastic along trajectories and was distributed mainly between one, two, and three motors, regardless of *N_m_*. We propose that this self-regulation allows our synthetic NPs to achieve persistent motion that is associated with major helicity. Such a helical motion might affect obstacle bypassing, which can influence active transport efficiency when facing the crowded environment of the cell.

## 1. Introduction

Active, motor-protein-mediated transport is crucial for the intracellular conveyance of a large variety of cargos in eukaryotes. Notably, microtubule-associated (MT-associated) motor proteins—dynein and kinesin—play a cardinal role, fueling a variety of vital biological processes [1,2,3]. While dynein is responsible for transport towards the cell center, members of the kinesin family are mostly responsible for transport towards the cell periphery [4,5,6,7]. The dynamic interplay between these two classes of motion orchestrates the subcellular arrangement of organelles, e.g., mitochondria [8] and Golgi complexes [1]. In addition, different types of viruses have evolved to harness the dynein machinery for the efficient targeting of the nucleus, where the infection process of the host cell occurs. HIV and herpes-simplex virus, for instance, express nuclear localization signal (NLS) peptides [7] that recruit dynein from the cytoplasm [9,10,11,12,13]. Likewise, adenoviruses use other ligands (e.g., hexon) to engage the active transport mechanisms [14,15]. Recently, preliminary theoretical work has been performed to mimic these viruses in rationally designed cargos that can be used for drug delivery applications [16].

While knowledge of single dynein motility has increased extensively in the past decade [17,18,19,20,21,22,23,24], much less is known about the collective motility of multiple motors that carry a single cargo. Several observations have led to the prevailing belief that native cargos are carried by more than one motor [25,26,27,28,29]. Studies of supercoiled DNA plasmids, enriched by NFκB receptors, showed robust nuclear temporal localization [30], supporting multiple dynein recruitment mediated by NLS binding. Complexes of fluorescently labeled single-strand DNA, such as the VirE2 protein of *Agrobacterium* and the VirE2 itself, contain putative NLS regions that can recruit dynein [31].

The collective behavior of motor proteins that carry a single cargo is often described in oversimplified terms, such as cooperative or agonistic behavior (where motors walk in unison) vs. antagonistic behavior (where motors interfere with each other’s walk) [32,33,34,35,36]. Nevertheless, this distinction is far from being definite. Moreover, recently, complex modes of motion have also been observed: in particular, a remarkable helical motion of dynein-coated beads, which demonstrated both right- and left-handed helices [37,38,39]. A theoretical description for the sideway motion of a single *yeast* dynein has recently been put forward [40] in qualitative accord with these experiments. Previous theoretical works modeled certain features of multi-motor motion [16,32,33,36,41,42,43,44,45,46,47,48,49,50,51].

In the absence of motor–motor elastic coupling, one finds (theoretically) an exponential increase of the run-length with an increasing number of cargo-associated motors [16,49,50]. However, motor–motor elastic coupling [48], combined with excluded volume interaction between motors, may strongly affect motion and lead to different characteristics from those of the single-motor behavior. Moreover, these theoretical studies do not account for essential features of dynein stepping, for example, they do not include the detailed locations of the dynein binding sites over the MT surface. In this spirit, as mentioned above, the *single motor* stepping was recently revisited to describe this two dimensional (2D) stepping of a single *yeast* dynein [40]. The 2D stepping model correctly accounts for the measured longitudinal step-size distribution and predicts a broad angular distribution of steps with a small right-handed bias. Elucidating the role of motor–motor coupling in realistic (biological) multi-motor complexes is crucial for understanding the behavior of actively transported native cargos.

Motivated by a previous theoretical work [16], we present here a combined experimental–theoretical research on rationally designed particles that can serve as a model particle for native multi-motor complexes, on the one hand, and allows a quantitative examination of their transport characteristics and their association with motor–motor coupling, on the other hand. Experimentally, our strategy is based on grafting a spherical nano-particle (NP) with a prescribed grafting density of Biotin-polyethylene-glycol-thiol (Biotin-PEG-thiol) molecules and end-linking a controlled fraction of the Biotin-PEG-thiols with a single NLS (Figure 1A). We expected, therefore, that under exposure to a cell extract (CE), the NLS peptide would first recruit an *α*-importin protein and then bind a *β*-importin protein, which, subsequently, would recruit dynein. Thus, the Biotin-PEG-thiol, which connects the dynein and the NP, serves as a spacer polymer with variable flexibility (depending on its contour length, i.e., molecular weight). By stretching, these spacers allow for motors originating from the sides of the NP to readily reach the MT surface (Figure 2A,B), thereby, increasing the number of motors participating in the transport. It follows that the structural properties of the NP, such as its Biotin-PEG-thiol grafting density, linked NLS fraction, Biotin-PEG-thiol contour length, NP size, and CE concentration, will, together, govern the number of dynein motors that bind to the MT simultaneously, thereby, potentially controlling the motility of the NP.

Our NPs are quite different from the cargo designs that have been previously reported [37,38,39]. First, previously reported cargos do not involve the native protein assembly for the recruitment of dynein from the cytoplasm. Second, their size is one order of magnitude larger than the NPs used in this study. (Note that the radius of a native cargo size is typically in the range of 20,100 nm, similar to the size of the NP used here). Importantly, in this radius range, the drag force on the NP is entirely negligible. (Using the NP translational Stokes drag coefficient, $\gamma_{t}=6\pi \eta R$, and a velocity of v=4 µm/s, leads to a drag force f= $\gamma_{t}v\sim 10^{-2}\;\mathrm{pN}$, for η=10 mPa s  as an upper bound for the CE viscosity, which is much smaller than the typical motor force of 4 pN.). This implies that the single motor will move at its free-load velocity and without a load-sharing effect [52]; such an effect will occur only for micron-sized NPs, for which the drag force becomes significant [38,53]. Third, the cargos used in previous studies [37,38] cannot control the mechanical coupling between motors. As such, they are less suitable for a fundamental experimental–theoretical study of a model multi-motor cargo, which could lead to an understanding of the motion of native cargos. We also note that, to the best of our knowledge, previous transport studies of dynein-recruiting cargos that use NLS ligands and are exposed to CEs have been performed only in 3-dimensional MT networks [7], which did not allow the characterization of the motion on individual MT tracks.

In this work, we performed—using our NP constructs—bead-motility assays with total internal reflection fluorescence (TIRF) microscopy. We used single-particle tracking algorithms to extract the trajectories of the NPs, and we measured several properties of these trajectories: (i) modes of motion, such as directional motion and hops between crossing MTs; (ii) run-lengths; (iii) run-times; and (iv) off-axis steps. To elucidate the factors that control NP motility, we modeled the active transport of the NP by including several competing processes, such as the binding to and the unbinding from the MT surface and the stepping kinetics of individual dynein motor proteins. These processes take place on the curved 2D microtubule surface and are influenced by both the elastic coupling between the motors (via the spacer polymers) and the excluded-volume interaction between motors. We used Monte-Carlo (MC) simulations to describe these dynamics. We highlight the unique features of the multi-motor carried NP by comparing between the experiment and theory-based simulations, which mutually support each other. Moreover, this comparison between the experimental and theoretical results provides insights into the mechanism underlying NP motion and, in particular, into the role of multiple motor action during transport.

## 2. Materials and Methods

### 2.1. Materials

#### 2.1.1. Preparation of Hela Cell Extracts

Hela CEs were prepared according to Fu et al. [54]. Briefly, Hela cells were grown in a DMEM medium supplemented with 10% fetal bovine serum (FBS) and 1% of penicillin-streptomycin at 37 °C and 5% CO_2_. The cells were detached by a trypsin-EDTA solution, washed with phosphate-buffered saline (PBS) (0.01 phosphate buffer, 0.0027M KCl, 0.137 M NaCl, pH 7.4), and pelleted for 5 min at room temperature (RT) and 500 g. The cells were then incubated on ice with lysis buffer (12 mM Pipes, 2 mM MgCl_2_, 1 mM EGTA, 0.4% Triton X-100, 10% protease inhibitor cocktail, pH 6.8) for 15 min. Finally, the lysates were cleared by centrifugation at 2700× *g* and then at 100,000× *g*, at 4 °C, for 10 min for each lysate. The CEs were diluted 10× with a lysis buffer without Triton and protease inhibitors, and the total protein mass concentration was determined by a Bradford assay, henceforth denoted by [CE]. Sucrose was added to the extracts (10% in mass), which were then flash-frozen in liquid nitrogen and stored at −80 °C. For the motility experiments, the extracts were diluted to the same total protein mass concentration of 3.4 mg/mL (systems I–IV), except for system V for which a [CE] of 6.8 mg/mL was used. The extracts were used within four weeks.

#### 2.1.2. Nuclear Localization Signal (NLS)

The NLS sequence (PKKKRKVED) originated from an SV40 T large antigen [55]. We used N-terminally bromine (Br-Ac) modified NLS. The bromine group was followed by a GGGG sequence (“raft”). To generate a fluorescently labeled NLS peptide, we covalently attached Tetramethylrhodamine (TAMRA) to the NLS raft. Except for the UV-Vis absorption experiments, in which we used TAMRA-NLS, all experiments were performed with the non-labeled NLS.

#### 2.1.3. Preparation of NPs

Bare, green-fluorescent microspheres (Bangs Labs, Indianapolis, IN, USA or Invitrogen, Waltham, MA, USA) were used to prepare the NPs. The total surface area of the bare NPs was constant throughout all the experiments, which guaranteed that the mean anchoring distance between PEG-NLS molecules was independent of the NP diameter. As a starting point, one can also use streptavidin-coated NPs (Bangs Labs, IN, USA) instead of bare microspheres. NP preparation started with the incubation in 10 µM Neutravidin (31,000, Thermo Fisher, Waltham, MA, USA) for 20 min at 25 °C. Excess Neutravidin was separated via dialysis. Next, the NPs were incubated for 20 min at 25 °C  in 10 mg/mL BSA to block the remaining uncoated regions on the NPs surface. Excess BSA was separated by centrifugation (6800× *g* for 8 min at 25 °C). All subsequent centrifugation steps were performed under the same conditions. The NPs were then incubated with 1 mM Biotin-PEG-thiol-5kDa (PG2-BNTH-5, NANOCS, Boston, MA, USA see Appendix A) for 30 min at 25 °C. Excess Biotin-PEG-thiol was separated by two cycles of centrifugation. Then, NLS peptides were covalently attached to the NPs via their bromine group, which reacts with the thiol group on the PEG molecule. The reaction was performed at 25 °C for 20 min. Excess NLS was separated by two cycles of centrifugation. Finally, the NPs were incubated with Hela CEs at 30 °C for 20 min, allowing for the recruitment of *α*- and *β*-importins, dynactin, and *mammalian* dynein motors to the surface of the NPs. The NPs were separated from excess CE by centrifugation, washed with BRB80 (80 mM  PIPES pH 6.8, 1 mM MgCl_2_, 1 mM EGTA) supplemented with 1 mM Mg-ATP, and centrifuged again. Finally, the NPs were resuspended in 60 µL BRB80 supplemented with 1 mM Mg-ATP and stored on ice until used. In the motility assay experiments, two different batches of CEs were used: the same CE was used in systems I and II, and another CE was used in systems III, IV, and V. To ensure full activity of the motility protein machinery, the NPs were used within 3 h.

### 2.2. Methods

#### 2.2.1. Cryo-Electron Microscopy (Cryo-TEM)

Vitrified specimens were prepared according to a standard procedure [56]. Briefly, 2.5 μL drops of an NP solution were applied to a copper grid coated with a perforated lacy carbon 300 mesh (Ted Pella Inc., Redding, CA, USA) and blotted with a filter paper to form a thin liquid film. The blotted samples were immediately plunged into liquid ethane at its freezing point (−183 °C) using an automatic plunge freezer (EM GP, Leica Microsystems GmbH,Vienna, Austria) and transferred into liquid nitrogen for storage. The samples were analyzed using an FEI Tecnai 12 G2 TEM at 120 kV with a Gatan cryo-holder, maintained at −180 °C. Images were recorded on a CCD camera (Gatan manufacturer, Pleasanton, CA, USA) at low-dose conditions.

#### 2.2.2. Dynamic Light Scattering and ζ-Potential Experiments

DLS and *ζ*-potential measurements were used to extract the NP hydrodynamic diameter$\;(D_{h}$) and charge at the various decoration steps. For these experiments, we used bare NPs with a 0.196 µm diameter. The measurements were performed on a Malvern NanoZS instrument (ZN-NanoSizer, Malvern, UK) operating with a 2 mW HeNe laser at a wavelength of 632.8 nm. The detection angle was 173° and 17° for the DLS and *ζ*-potential measurements, respectively. All measurements were conducted at 25 ± 0.05 °C; for the analysis of $D_{h}\;$ and *ζ*-potential, the viscosity was taken to be that of water (0.8872 cP). The intensity size (hydrodynamic diameter) distribution was extracted from the intensity auto-correlation function, which was calculated using an ALV/LSE 5003 correlator over a time window of 30 s (10 runs of 3 s) using the software CONTIN. Each measurement was repeated three times. The value of $D_{h}\;$ was averaged over three independent experiments; error bars indicate the standard deviations for these three experiments. For *ζ*-potential measurements, the solution was transferred to a U-tube cuvette (DTS1070, Malvern, UK), which was operated in automatic mode. The electrophoretic mobility of the NPs was measured, from which the *ζ*-potential value was determined by applying the Henry equation [57]. The *ζ*-potential values were averaged over three independent experiments with 30 runs per experiment; error bars indicate the standard deviations for these three experiments.

#### 2.2.3. UV-Vis Absorption Experiments

UV-Vis absorption experiments were used to determine the mean number of bound PEG-NLS molecules per NP, 〈N〉, from which the mean anchoring distance between adjacent PEG-NLS molecules, $\xi^{*}$, was deduced. These experiments were conducted at a wavelength of 553 nm, using a fluorescently labeled NLS (TAMRA-NLS). Biotin-PEG-thiol-grafted NPs were incubated with increasing amounts of TAMRA-NLS and the absorption of the remaining TAMRA-NLS molecules that did not bind to the NPs (i.e., supernatant) was measured. The number of TAMRA-NLS molecules that *did* bind to the NPs was then deduced (see details in Appendix A). An extinction coefficient $\varepsilon_{553}=0.0639\left[\frac{1}{\mu M\;\mathrm{cm}}\right]$, which we determined experimentally (not shown), was used for that purpose. The values of 〈N〉 and $\xi^{*}$ were averaged over three independent experiments; error bars indicate the standard deviations for these three experiments.

### 2.3. Western Blotting

Western blot (WB) was used to confirm the recruitment of importins and dynein motors by the PEG-NLS coated NPs. These experiments employed NPs of 0.196 µm in diameter, [NLS] of 7 μM, and [CE] of 3.4 mg/mL. The NPs were prepared as detailed above, then pelleted at 6800× g for 10 min at 25 °C, resuspended in a 1× Laemmli Sample Buffer, and boiled for 5 min to promote the detachment of the bound proteins. The proteins were separated by electrophoresis using a 12% agarose gel and transferred to a nitrocellulose membrane. The membrane was incubated for 1 h in a blocking buffer of PBST (PBS supplemented with 0.1 *v*/*v* % Tween) and 10% (*v*/*w*) dry skim milk (Sigma-Aldrich, St. Louis, MO, USA). The membrane was washed three times with PBST for 5 min and then incubated for 1 h at 25 °C with anti-dynein (sc-13524), anti-karyoprotein α2 (sc-55538), anti-dynactin (sc-135890), and anti-kinesin-1 (sc-133184) (Santa Cruz Biotechnology, Dallas, TX, USA). All primary antibodies were diluted 1:200 (*v*/*v*) in a blocking buffer prior to use, except anti-dynactin, which was diluted to 1:50 (*v*/*v*). Next, the membrane was washed three times with PBST and incubated for 1 h at 25 °C with an anti-mouse HRP conjugated with a secondary antibody (sc-2005, Santa Cruz Biotechnology, Dallas, TX, USA) diluted 1:10,000 (*v*/*v*) in PBST supplemented with 0.5% (*v*/*w*) skim milk. To finalize the procedure, the membrane was washed three times with PBST and incubated with an ECL Western blot reagent (1,705,060, Bio-Rad, Hercules, CA, USA) for 5 min in the dark. Images were collected by chemiluminescence, using the Fusion FX imaging system (Vilber Lourmat, Collégien, France).

### 2.4. Motility Assay Experiments

Chamber preparation. Flow cells were prepared using a glass slide and a glass coverslip (washed with deionized water, 70% EtOH, and dried using nitrogen gas), and two stripes of warm Parafilm were placed in between them to form a channel with a width of a few millimeters. The chamber was incubated with one volume of 0.1 mg/mL Biotinylated Casein (prepared by biotinylation of k-Casein (Sigma-Aldrich, St. Louis, MO, USA) using EZ-Link (21,336, Thermo Fisher, Waltham, MA, USA) for 5 min at 23 °C and washed with three volumes of BRB80. The chamber was then incubated with one volume of 1 mg/mL Neutravidine (31,000, Thermo Fisher, Waltham, MA, USA) for 5 min at 23 °C and washed with three volumes of BRB80.

### 2.5. Preparation of Microtubules

Tubulin from a porcine brain, purified via three polymerization/depolymerization cycles [58], was flash-frozen in liquid nitrogen and kept at −80 °C until used. Biotin-fluorescently labeled microtubules were prepared according to the protocol described in the work of Gell et al. [59]. A tubulin mix, containing 51 µM tubulin, 6 µM Biotin tubulin (T333P, Cytoskeleton, Denver, CO, USA), 3 µM Rhodamine tubulin (TL590M, Cytoskeleton, Denver, CO, USA), and 3 mM GMPCPP (NU-405S, Jena Bioscience, Jena, Germany), was mixed on ice, divided, flash-frozen in liquid nitrogen, and kept at −80 °C until used. Prior to the mix preparation, the tubulin was thawed and kept on ice for 5 min and then centrifuged for 10 min at 126,000× *g*, 4 °C. The supernatant was kept on ice and its concentration was determined by absorbance at 280 nm ($\varepsilon_{280}=115,000\left[\frac{1}{M\;\mathrm{cm}}\right]$). For microtubule (MT) assembly, an aliquot of the tubulin mix was thawed, diluted with BRB80 to a final concentration of 4 µM, and incubated for 3 h at 37 °C to promote MT assembly. For motility assay experiments, the MTs were diluted to 40 nM with a warm (37 °C) wash buffer (WshB) (BRB80, containing 0.02 mM  paclitaxel, 15 mM glucose, 0.1 mg/mL glucose oxidase, 0.02 mg/mL catalase, and 50 mM DTT) and used on the same day. The MTs were kept at 23 °C until used.

### 2.6. Formation of Marked-End Microtubules

Marked-end microtubules were prepared by using bright fluorescent MT seeds, which served as the nucleation sites for the polymerization of dim fluorescent, N-Ethylmaleimide (NEM)-modified tubulin [60,61]. The NEM-modified tubulin was used to inhibit the MT seed minus-end assembly, and it was prepared by mixing 100 µM tubulin with 1 mM NEM (Sigma-Aldrich, St. Louis, MO, USA) and 0.5 mM GMPCPP in BRB80 and placing the solution on ice for 10 min. The reaction was quenched by adding 8 mM β-mercaptoethanol. The NEM-tubulin mix was incubated on ice for an additional 10 min, and then a tubulin mix, containing 3.2 µM tubulin, 4 µM Rhodamine tubulin, 0.8 µM Biotin tubulin, and 1 mM GMPCPP in BRB80, was incubated for 15 min at 37 °C to produce short “bright” MT seeds. One volume of bright MT seeds mixed with seven volumes of a “dim” tubulin mix (containing 3.6 µM tubulin, 0.4 µM Rhodamine tubulin, 0.8 µM Biotin tubulin, 3.2 µM NEM-modified tubulin, and 1 mM GMPCPP in BRB80) was incubated at 37 °C for 1 h to promote polymerization. The marked-end MTs were diluted to 40 nM with warm (37 °C) WshB and used on the same day. The MTs were kept at 23 °C until used.

### 2.7. Motility Assay Experiments

For these experiments, 40 nM of MTs were introduced in the flow cell and incubated for 10 min at 23 °C. The chamber was washed with three volumes of WshB to remove unbound MTs. Prior to the assay, the NPs were pelleted by centrifugation at 6800× *g* for 8 min at 4 °C, resuspended in 60 µL motility buffer (BRB80 containing 0.02 mM Paclitaxel, 15 mM glucose, 0.1 mg/mL glucose oxidase, 0.02 mg/mL  catalase, 50 mM DTT, 0.1% methyl cellulose 4000 cP, and 10 mM Mg-ATP), and incubated on ice for 5 min. Prior to insertion in the flow cell, the NPs were brought to 23 °C. The motility assays were performed with bare NPs of 20 nm in radius.

### 2.8. NP Imaging Using Total Internal Reflection Fluorescence Microscope (TIRFM)

We followed the motion of the NPs by TIRF microscopy using an automated TIRF multicolor system integrated on an inverted Leica DM6000B microscope (Leica Microsystems GmbH, Wetzlar, Germany). Imaging was performed using a HCX Plan Apo 1.47 N.A. 100× TIRFM oil immersion objective and a triple-band Laser Line Violet Blue Green (VBG) emission filter (Leica Microsystems GmbH, Wetzlar, Germany). The samples were excited by TIRF illumination using 10 mW 488 nm and 20 mW 561 nm solid-state lasers. The images were captured using an Andor DU-897 EMCCD camera (Oxford Instruments, Abingdon, UK). The imaging system produced images of the NPs with apparent radii of ~3 pixels (pixel size is 230 nm), which allowed the extraction of the center-of-mass coordinates of the NPs, NP(X,Y), and the center-of-mass coordinates of the MTs, ${\left(X,Y\right)}_{\mathrm{MT}}$, with a sub-pixel resolution and a 10 nm scale precision (see below).

### 2.9. Particle Tracking and Data Analysis

We used the interactive data language (IDL) multi-particle tracking method [62,63], implemented in MATLAB, to automatically detect distinct NP trajectories. Using this method, we extracted the 2D center-of-mass coordinates (X(t),Y(t)) per time point t (frame) of the individual NPs by fitting the NP fluorescence profiles along the X and Y directions using a Gaussian function, from which we determined the individual NP trajectory over time, $\mathrm{NP}{\left(X,Y\right)}_{t}$. We conducted an initial (automatic) filtering, in which we selected trajectories that included at least three steps. Next, we performed a more delicate (manual) filtering, in which we excluded NPs or NP trajectories that fell under one or more of the following categories: (i) the NP shape was not symmetric or too large, which implies NP aggregation; (ii) the NP moved in a region where the density of MTs was too dense, and thus, it was impossible to discern between individual MT tracks; and (iii) the NP moved only for a short period and then got stuck along the MT track for a long period. For each NP, the overall run-time and run-length were calculated from the accumulated travel time and distance, respectively. In some cases, where the NPs moved continuously between crossing MT tracks, we extracted the trajectory on each MT track separately. For these NPs, the overall run-time and run-length correspond to the accumulated traveled time/distance of all individual trajectories.

The NP velocity v was calculated by taking the center-of-mass position of the NP at times t, ${\left(X(t),Y(t)\right)}_{\mathrm{NP}}$ and t+Δt ${\left(X\left(t+\Delta t\right),Y\left(t+\Delta t\right)\right)}_{\mathrm{NP}}$, and dividing it by Δt, that is $v=\frac{\Delta r_{\mathrm{NP}}}{\Delta t}=\frac{{\left(X\left(t+\Delta t\right)-X\left(t\right),Y\left(t+\Delta t\right)-Y\left(t\right)\right)}_{\mathrm{NP}}}{\Delta t}$, where X and Y refer to lab frame Cartesian coordinates. The longitudinal velocity, $v_{x}$, was calculated by projecting the NP velocity v  onto the MT direction (by definition, a positive direction points towards the MT minus-end), i.e., the unit vector of the MT (along its long axis), ${\hat{u}}_{\mathrm{MT}}=\frac{\Delta r_{\mathrm{MT}}}{\left|\Delta r_{\mathrm{MT}}\right|}$ (Appendix A), where $\Delta r_{\mathrm{MT}}={\left(X_{f}-X_{i},Y_{f}-Y_{i}\right)}_{\mathrm{MT}}$. The indexes i and f refer to the initial and final time points of the NP trajectory, and ${\left(X_{i},Y_{i}\right)}_{\mathrm{MT}}$ and ${\left(X_{f},Y_{f}\right)}_{\mathrm{MT}}$ are the corresponding MT center-of-mass coordinates, which are extracted by fitting the cross-section intensity profiles of the MTs using a Gaussian function. Thus, the longitudinal velocity $v_{x}=v\cdot {\hat{u}}_{\mathrm{MT}}=\frac{\Delta x}{\Delta t}=\frac{{\left(X\left(t+\Delta t\right)-X\left(t\right),Y\left(t+\Delta t\right)-Y\left(t\right)\right)}_{\mathrm{NP}}}{\Delta t}\cdot \frac{{\left(X_{f}-X_{i},Y_{f}-Y_{i}\right)}_{\mathrm{MT}}}{\left|\Delta r_{\mathrm{MT}}\right|}$, where $\left|\Delta r_{\mathrm{MT}}\right|=\sqrt{{\left(X_{f}-X_{i}\right)}^{2}+{\left(Y_{f}-Y_{i}\right)}^{2}}$. Positive $v_{x}$ corresponds to an NP step Δx in the direction of the MT minus-end, whereas negative values of $v_{x}$ refer to an NP step towards the MT plus-end (Figure 2C and Appendix A).

Similarly, the transverse motion, $v_{y},\;$ was determined by projecting the NP velocity v  onto the normal unit vector of the MT, ${\hat{n}}_{\mathrm{MT}}$. Thus, the transverse velocity $v_{y}=v\cdot {\hat{n}}_{\mathrm{MT}}$ and the corresponding transverse step is given by $\Delta y=v_{y}\cdot \Delta t.$ Note that, for each MT or MT unit vectors ${\hat{u}}_{\mathrm{MT}}$, there are two opposite normal unit vectors that can be assigned, ${\hat{n}}_{1,\mathrm{MT}}$ and ${\hat{n}}_{2,\mathrm{MT}}$, which account for positive and negative NP transverse motions, respectively (Appendix A). In this paper, we defined the right-handed transverse motion as positive and the left-handed transverse motion as negative (see Figure 2C for the definition of spatial orientation).

Next, we filter unrealistic velocity values as follows: First, we filter out absolute velocity values, $\upsilon =\sqrt{\upsilon_{x}^{2}+\upsilon_{y}^{2}}$, exceeding $4000\;\frac{\mathrm{nm}}{s}$, which represents the very rare—and not very plausible—case where the dynein steps 100 consecutive 40 nm sized steps. Second, we filter out unrealistic transverse steps. To this end, we set a maximum size of a transverse step, $\Delta y=2\times R_{\mathrm{orb}}=2\times 153\;\mathrm{nm}$, which corresponds to the scenario where the NP moves half a circle along the MT perimeter, from left to right or vice versa. A transverse step that is larger than the said limit is omitted.

We use the mean value of transverse steps,⟨ Δy⟩, to extract the mean angular step, ⟨Δϕ⟩, and its error, $\Delta \phi_{e}$, and the mean angular velocity, ⟨ω⟩, and its error, $\omega_{e}$, as follows:(1)$$\langle \Delta \phi \rangle \pm \Delta \phi_{e}=\arcsin\left(\frac{\left\langle \Delta y\right\rangle }{R_{\mathrm{orb}}}\right)\pm \frac{\left\langle \Delta y\right\rangle }{R_{\mathrm{orb}}}\times \frac{1}{\sqrt{1-{\left(\frac{\mathrm{SEM}\left(\Delta y\right)}{R_{\mathrm{orb}}}\right)}^{2}}}$$
(2)$$\langle \omega \rangle \pm \omega_{e}=\frac{\left\langle \Delta \phi \right\rangle }{\Delta t}\pm \frac{\Delta \phi_{e}}{\Delta t}$$
where SEM(Δy) is the standard error of the mean (SEM) value of Δy, and $R_{\mathrm{orb}}$ is the approximated distance between the MT and NP center (Figure 2C and Figure 6A), which reads:(3)$$R_{\mathrm{orb}}=R_{\mathrm{MT}}+R_{N-\mathrm{NP}}+C_{L}+d_{\mathrm{dynein}}+d_{\mathrm{Importins}}≅153\;\mathrm{nm}$$
where$\;R_{\mathrm{MT}}=12.5\;\mathrm{nm}$ is the MT radius, $C_{L}=40.8\;\mathrm{nm}$ is the $M_{w}=5\;\mathrm{kDa}$ PEG polymer contour length (see Appendix A), $d_{\mathrm{dynein}}=45\;\mathrm{nm}$ is the dynein characteristic dimension, and $d_{\mathrm{Importins}}=15\;\mathrm{nm}$ is the dimension of the α- and β-importin complex [64,65,66,67,68] (see Appendix A). $R_{N-\mathrm{NP}}=40\;\mathrm{nm}$ corresponds to the largest Neutravidin-coated NP radius in the sample. The NP diameters were extracted from cryo-TEM images (data not shown). This sets the upper limit of accessible $R_{\mathrm{orb}}$.

Next, we calculate the mean helical pitch, $\left\langle \bar{H}\right\rangle$, and its corresponding error, ${\bar{H}}_{e}$, by
(4)$$\langle \bar{H}\rangle \pm {\bar{H}}_{e}=\frac{2\pi }{\left|\langle \omega \rangle \right|}\left|\langle \upsilon_{x}\rangle \right|\;\pm \;\sqrt{{\left(\;\frac{2\pi \;}{\left\langle \omega \right\rangle }\times \mathrm{SEM}\left(\upsilon_{x}\right)\right)}^{2}+{\left(\frac{2\pi \times \langle \upsilon_{x}\rangle }{{\left\langle \omega \right\rangle }^{2}}\times \omega_{e}\right)}^{2}}$$
where $\mathrm{SEM}\left(\upsilon_{x}\right)$ is the standard error of the mean value of $\upsilon_{x}$. Similarly, we can estimate the mean angular velocities for right- (⟨ω⟩|ω>0) and left- (⟨ω⟩|ω<0) handed motions and the mean angular velocity for separated minus-end directed ($\langle \omega \rangle |\upsilon_{x}>0$) and plus-end directed ($\langle \omega \rangle |\upsilon_{x}<0$) motions (see Appendix A).

## 3. Results

Below, our analysis of the experimental and model-simulation results will address the following parameters: (i) the mean anchoring distance between neighboring PEG-NLS molecules $\xi^{*}$; (ii) the mean anchoring distance between the PEG-NLS-αβ-dynein (αβ refers to the α- and β-importin complex) ξ; (iii) the number of motors, $N_{m}$, that are anchored to the surface of the NP; and (iv) the number of MT-bound motors, $M_{B},$ that participate in the motion at a given time. The mean number of NP-bound motors is related to ξ via $\langle N_{m}\rangle =\frac{4\pi R^{2}}{\xi^{2}}$, where R is the radius of the NP.

### 3.1. NP Preparation and Analysis

The NP synthesis process entails several consecutive steps where, at each step, a single component is added (Figure 1A). Bare NPs, saturated with Neutravidin, were conjugated with Biotin-PEG-thiol spacers and then incubated for a short time in a solution containing an SV40 T large antigen NLS peptide at various concentrations.

The resulting NPs are characterized (Appendix A) by using cryo-TEM (Appendix A), DLS, ζ-potential, and UV-Vis absorption isotherm, yielding the surface density and the mean number of grafted Biotin-PEG-thiols, ⟨N⟩, which are end-conjugated by NLS (PEG-NLS) (Appendix A). The latter can be transformed into a mean anchoring distance between neighboring PEG-NLS molecules, $\xi^{*}$. The resulting anchoring distance, $\xi^{*},$ is depicted in Figure 1B against [NLS], showing that the incubation of NPs in higher [NLS] solutions leads to a shorter distance between adjacent PEG-NLSs until it reaches saturation above [NLS] = 3 µM. Note that the theoretical value of the (free-polymer) gyration radius is $R_{g}=2.27\;\mathrm{nm}$ (for a Biotin-PEG-thiol of $M_{w}=5\;\mathrm{kDa};$ see Appendix A [69,70,71,72,73,74,75]). Thus, since the diameter of Neutravidin is about 5 nm [76], and since the Neutravidins are closely packed on the NP surface, we may conclude that the anchored Biotin-PEG-thiol molecules are effectively in the so-called “mushroom regime” [77], $\xi^{*}>R_{g}$, wherein the polymer chains appear as isolated “mushrooms” (i.e., random coils of radius $\sim R_{g}$) on top of the NP surface.

Next, NPs are incubated in Hela CEs, thus, allowing the recruitments of *α*- and *β*- importins, dynactin, and *mammalian* dynein association [54]; the recruitments of the importins, dynactin (which is required for the proper functioning of dynein [78,79]), and *mammalian* dynein are verified by WB (see Materials and Methods). Figure 1Ca shows the WB image after exposure to the CE, demonstrating antibody-specific binding to *α*2-importin. Figure 1Cb shows a WB image demonstrating dynein recruitment, and Figure 1Cc shows a WB image demonstrating dynactin recruitment. Group 1 refers to NPs coated with Biotin-PEG-thiol incubated in CE; group 2 refers to PEG-NLS coated NPs, which were also incubated in CE; and group 3 refers to a CE without NPs. Note that although *β*-importin binding was not verified directly, dynein recruitment requires *β*-importin binding, suggesting that, in the absence of NLS, the dynein machinery is not recruited to the NP surface.

An important question regarding our NP construct is whether it allows the non-specific binding of kinesin motors, which will influence the motility characteristics. To discard this possibility, we performed WB for kinesin-1 binding to the NP (Figure 1D). The WB image does not show kinesin-1 binding to the PEG-NLS coated NPs (although it remains possible that kinesin-1 binding occurs below the WB detection threshold). This finding is expected since the bare NP is covered with a first layer of Neutravidin, a second layer of BSA, and a third layer of Biotin-PEG-thiol, which, together, are expected to efficiently passivate the NP surface.

### 3.2. NP Motility Assays

The NPs were washed from the excess cell extract, incubated in an ATP solution, and then injected into a flow cell in which MTs were adsorbed and immobilized on a glass surface. We investigated the different modes of motion of the NPs under different [NLS] but identical [CE], as defined in Table 1. Thus, we investigated four systems, wherein extracts from an identical batch were used in systems I and II and extracts from another identical batch were used in systems III and IV. Since individual protein concentrations may vary from one CE to another (although the total mass protein concentration, [CE], that we used was identical; see Material and Methods), for the sake of prudence, we compared the results of the motility assays only within each pair: I vs. II and III vs. IV. We also examined another system, referred to as system V, whose CE batch was identical to that of systems III and IV but whose [CE] was twice as high; however, we do not discuss this system below since its motility is strongly reduced, presumably due to molecular crowding effects (see Appendix A). To follow the motion of the NPs, we used TIRFM (e.g., see Movie 1 (system II) and Movie 2 (system III)). First, we used marked minus-end MTs to confirm the expected correlation between the polarity of the MT and the motional direction of the NP, as dictated by the dynein bias to move towards the minus-end (Figure 3A,B). Next, we used a standard particle tracking algorithm to extract the center-of-mass (CM) position of the NP with a 10 nm-scale precision (see Materials and Methods), which yielded individual time-dependent trajectories. Notably, although several NPs can potentially move on the same MT track simultaneously, we mostly observed only a single NP at a time. Notwithstanding, to prevent possible interference from nearby NPs (e.g., obstruction), we purposely chose to analyze only NPs that moved alone on a single MT.

To analyze their motion, we differentiated the trajectories of the NPs into plus-end and minus-end directed motion, with respect to the movement along the long-axis of the MT (henceforth defined as “longitudinal motion”), see Figure 2C. We detected three major modes of motion (Figure 3 and Table 1): (i) NPs that moved continuously towards the MT minus-end (Figure 3A,B); (ii) NPs that reached the end of the MT after exhibiting continuous minus-end directed motion, followed by a single backward step (i.e., towards the MT plus-end) and detachment (Figure 3C,D). In some cases (not shown), the NP remained immobile for some time before detaching; and (iii) NPs that traversed between crossing MTs (Figure 3E,F, Movie 3). Importantly, trajectories that involve traversing between crossing MT tracks have been previously observed by Ross et al. in single-motor assays [80,81]; however, the simultaneous binding of two dynein motors to the NP should enhance the frequency of such traversing events with the help of an intermediate configuration where one motor binds to one track and the second motor binds to the other track.

For each CE batch, we tested how [NLS] affects NP motion. Table 1 summarizes the estimated values of the mean anchoring distance $\xi^{*}$ of PEG-NLS for each of the four systems, which we deduced from the mean number of grafted PEG-NLS, ⟨N⟩ (see Appendix A). Note that $\xi^{*}$ serves as an estimated lower bound for the mean anchoring distance ξ of the PEG-NLS-αβ-dynein, such that ξ should approach $\xi^{*}\;$(from above) as the concentration of associated protein concentration (α-  and β-importins, dynein, dynactin, etc.) in the CE increases (ξ was not directly measured). Moreover, given the expected sub-millimolar concentrations of dynein [82], we assume the system is much below the saturation of the dynein binding “isotherm”, such that the increase of CE proteins (and, accordingly, the dynein bulk concentration) is likely to increase dynein surface concentration, resulting in a decrease of ξ. Thus, we expect systems I and II, which belong to the same CE batch and have the same [CE], will demonstrate the same value of $\xi /\xi^{*}$ or, equivalently, the same value $\frac{\left\langle N\right\rangle }{\left\langle N_{m}\right\rangle }={\left(\frac{\xi }{\xi^{*}}\right)}^{2}$; a similar argument applies to systems III and IV. The value of $\xi^{*}$ determines the mean number of PEG-NLS per NP via $\langle N\rangle =\frac{4\pi R^{2}}{\xi^{*}{}^{2}}$, which implies that ⟨N⟩ varies between 5 and 37 for [NLS] that varies between 0.025 and 0.3 μM (Table 1 and Appendix A).

#### 3.2.1. Longitudinal Velocity 

The mean properties of the NP trajectories are shown in Figure 4 and are summarized in Appendix A for each system. Except for system I, we found that the longitudinal velocities of the NPs significantly decreased compared with those of a single motor. Comparing each of the pairs (I to II and III to IV), we can deduce that the number of NP-attached motors increased such that $N_{m}$ was higher in system II than in system I and in system IV than in system III. Assuming that the mean number of motors participating in the transport $\langle M_{B}\rangle$ increases with $N_{m}$ (as shown by our theoretical predictions in Section 3.4, below) suggests that NP velocities decrease with increasing $\langle M_{B}\rangle$, presumably due to inter-motor interactions. We emphasize that the regime of transport corresponds to *vanishing drag* due to the nanometer size of the NPs, implying that a load-sharing effect should not be present whatsoever; such a reduction of the velocity along with the increase in the number of participating motors is commonly observed in standard motility assays of MTs moving on kinesin/dynein-decorated glass surface [83,84]. Examining the average NP run-time, $\left\langle \tau_{p}\right\rangle$, and run-length (along the MT symmetry axis), ⟨λ⟩, for the different systems (Figure 4B and Appendix A) revealed that $\left\langle \tau_{p}\right\rangle$ increases with increasing [NLS] or [CE], leading to a decrease in ξ or an increase of $N_{m}$, consistent with the anticipated increase in the mean number of participating motors,$\;\langle M_{B}\rangle$ (*c.f.*
Section 3.4). Note, however, that since $\left\langle \lambda \rangle =\langle \upsilon_{x}\times \tau_{p}\right\rangle$, the dependence of the run-length on [NLS] and [CE] is non-monotonous and shows a maximum for system IV (Appendix A).

Notably, about 90% of the steps in our experiments were minus-end directed ($\upsilon_{x}>0$); Appendix A, and Appendix A show the distributions of the *temporal absolute* velocity, υ, longitudinal velocity, $\upsilon_{x}$, and transverse velocity, $\upsilon_{y}$, respectively. To further characterize the NP motion, as shown in Figure 4C,D and in Appendix A, we extracted the mean values of the longitudinal velocity (minus-end directed motion, $\upsilon_{x}>0$, and plus-end directed motion,$\;\upsilon_{x}<0$) and of the transverse velocity (left, $\upsilon_{y}<0$, and right, $\upsilon_{y}>0$). As [NLS] increases, which implies a decrease in ξ and an increase in the number of NP bound motors,$\;\langle N_{m}\rangle =\frac{4\pi R^{2}}{\xi^{2}}$, the corresponding absolute and longitudinal mean velocities decrease. Conversely, the probability of either the plus-end or the minus-end directed motions do not vary greatly (Appendix A). Notably, as the [NLS] increases (i.e., moving from system I to II and from system III to IV), the plus-end directed longitudinal velocity, $\langle \upsilon_{x}\rangle \;|\;\upsilon_{x}<0$, decreases to a much greater extent than the minus-end directed longitudinal velocity $\langle \upsilon_{x}\rangle \;|\;\upsilon_{x}>0$ (Figure 4C).

The observed variability in the properties of motility between system II and system III is relatively small; indeed, as can be clearly seen in Figure 4B, the SEMs of $\tau_{p}$ and λ overlap between systems II and III. Thus, the relatively small differences in the properties of motility between the two systems can be rationalized by the expected small difference in the protein composition of the CE.

We next argue that NP motility properties are only sensitive to the number of NP-associated motors, $N_{m}$. Consider the two observables $\langle \upsilon_{x}\rangle \;|\;\upsilon_{x}>0$ and$\;\tau_{p}$. In Figure 4E, we combine the data from the four different systems, I–IV, which vary either in PEG-NLS surface coverage, CE preparation, or both. In the main panel, we plot, for each NP, its $\langle \upsilon_{x}\rangle \;|\;\upsilon_{x}>0$ against $\tau_{p}$ as a separate data point. In the inset, we average the overall NPs to obtain a single functional dependence. It is clearly visible that there is some overlap of data between the different systems. A plausible explanation is that $N_{m}$ is effectively identical for overlapping NPs, regardless of their associated system—a hypothesis that will be further tested against the model simulations, as presented below.

#### 3.2.2. Angular/Transverse Motion

As discussed above, while most studies of dynein motors assume a purely longitudinal motion, more recent studies discovered rich transverse dynamics [37,38,52,85]. Therefore, we analyzed the NP trajectories for transverse motion as well. We first calculated the estimated transverse velocity,$\;\upsilon_{y}$, of the NPs (Figure 4D) and found that it was significantly lower than the longitudinal velocity.

To gain further insight into the transverse dynamics, we mapped the transverse velocity into an angular velocity, $\omega =\frac{\Delta \phi }{\Delta t}$ (where Δϕ is the angular increment of a single time step, Δt), as shown in Figure 5, Appendix A (see Materials and Methods for details). We assumed an ideal angular motion with constant orbital radius, $R_{\mathrm{orb}}=153\;\mathrm{nm}$ (Figure 2C), which we estimated by considering the molecular dimensions of the NP and its associated ligands (see Materials and Methods and Appendix A). The angular motion, together with the longitudinal motion, imply that if the MT had been elevated above the surface, the NP would have performed a helical motion around the symmetry axis of the MT. Indeed, one of the NPs showed such a helical motion, which was made possible by the (unintended) elevation of the specific MT on which it moved (see Movie 2, pink arrow). Considering the mean angular velocity (associated with Figure 5A) as a representative number, we estimated the mean pitch size of the assumed helical trajectory by using $\langle \bar{H}\rangle =2\pi \left|\left\langle v_{x}\right\rangle \right|/\left|\langle \omega \rangle \right|\;$(see Materials and Methods and Appendix A), which yielded the results shown in Figure 5B and Appendix A.

The trajectories of a single NP taken from the theoretical model simulations (presented in the next section) show large angular motion “fluctuations” of varying magnitudes (Movies 4–7 and Figure 6). To analyze these fluctuations, we split the angular motion into right- and left-handed motions (Appendix A; see Materials and Methods for details), yielding the angular velocities shown in Appendix A. We found that, indeed, the right- (⟨ω⟩|ω>0) and left- (⟨ω⟩|ω<0) handed motions occurred at similar angular velocities in all examined systems. In the Appendix A, we provide the correlations between the positive and negative longitudinal steps, on the one hand, and the positive and negative angular steps, on the other hand (Appendix A and Appendix A). However, due to the large inherent fluctuations associated with the angular motion, we cannot come to any conclusions about such correlations. The possibility of such correlations, between longitudinal and angular steps, is further examined by the model simulations, as detailed below.

### 3.3. Model and Simulations

To explain the experimental findings and to gain profound knowledge regarding NP motion and behavior, we constructed a model for the active transport of the NP. Since previous work on multi-motor complexes addressed a single-motor motion on a 1D microtubule track [32,33,36,42,51], it cannot describe the motion of the NP around the 2D MT surface. The single-motor stepping was recently revisited to describe the 2D stepping of a single *yeast* dynein on the microtubule surface; this stepping model correctly accounts for the measured longitudinal step-size distribution and, moreover, predicts a broad angular distribution of steps (of a single motor) with a small right-handed bias [40]. More recently, Elshenawy et al. [38] studied the single *mammalian* dynein stepping kinetics, providing the longitudinal and transverse step-size distributions. Using the raw data of Yildiz and co-workers [38,53], we adjusted our *yeast* dynein model [40] to describe a single *mammalian* dynein (see Appendix A).

Our NP multi-motor model is described in Appendix A and Appendix A. The model entails an NP on which polymers with an identical contour length and a fixed density, corresponding to a prescribed mean spacing $\xi^{*}$, are grafted at random positions. Each free polymer-end is assigned with a dynein motor, such that $\xi =\xi^{*},$ by definition. Thus,$\;\xi =\sqrt{4\pi \;}R/\sqrt{N_{m}}$, where R is the NP radius and $N_{m}$ is the number of motors that are anchored to the NP surface. Dynein is assumed to bind to and unbind from the MT surface, processes that compete with its stepping kinetics (see Appendix A). Note that the term “unbinding” refers to the full detachment of the motor from the MT, which implies that its two microtubule binding domains (MTBDs) become disconnected from it, such that the inverse of the single motor unbinding rate defines its processivity time. Likewise, “binding” refers to the attachment of a motor to the MT surface from the bulk solution. The motion of the motors on the 2D curved microtubule surface is influenced by the elastic coupling between motors (via the spacer polymers)—a coupling that influences both the motor step vector distribution and the binding–unbinding kinetics. In addition, we account for the excluded-volume interaction between motors that prevents them from stepping over each other; however, we do not account for the excluded volume interaction between different spacer polymers, which, we believe, is negligible (Appendix A [86,87,88]). Consistent with the nanometer scale of the NP, the drag force on the NP is negligible (see Section 1). We used Monte-Carlo (MC) simulations to describe the different competing processes, while readjusting the NP center-of-mass and rotational angle after each MC step to achieve mechanical force balance—processes that are orders of magnitude faster than the binding/unbinding and stepping processes (Appendix A). The model allows the investigation of the different microscopic internal states between which the NP fluctuates in time, in particular the number of MT-bound motors, $M_{B},$ that participate in the motion at a given instance.

### 3.4. Model Simulation Results

Results were obtained for various NP configurations, which are characterized by two parameters: R and $N_{m}$. Below, we discuss an NP with radius R=20 nm and a number of motors ranging between $N_{m}=1$ to $N_{m}=13$. For each $\left(R=20\;\mathrm{nm},\;N_{m}\right),$ we ran the simulations for a few hundred (identical) particles to obtain high statistical accuracy. Note that due to the very small NP drag coefficient (Appendix A), we used $N_{m}=1$ as a representative of the single-motor case. Figure 7 and Figure 9, and Appendix A, describe the motility characteristics, on a time scale of 0.27 s, of each NP configuration, while Figure 8, Figure 9 and Figure 10 and Appendix A complement these with analyses on the timescale of a single MC step ∼1 ms, which is two orders of magnitude shorter. Figure 6 and Movies 4–7 show selected trajectories of single-motor, three-motor, seven-motor, and thirteen-motor NPs, which demonstrate rich, fluctuating, helical motions. Below, we investigate the influence of the number of motors on such behavior.

#### 3.4.1. Longitudinal Velocity 

Figure 7A shows the mean longitudinal velocity, $\left\langle \upsilon_{x}\right\rangle$, against $N_{m}$. We found a suppression of $\left\langle \upsilon_{x}\right\rangle$ from the single motor velocity (833±4 nm/s) as $N_{m}$ increases, which is a clear signature of motor–motor coupling in the absence of any load-sharing effect (due to the vanishing drag). The corresponding standard deviation of the mean (STD) also varies with $N_{m}$; the maximal STD appears at$\;N_{m}=4\;$(corresponding to ξ=35.4 nm), whereas the minimal STD corresponds to the single-motor case. In addition, as seen in Figure 7B, the characteristic run-time, $\left\langle \tau_{p}\right\rangle$, and longitudinal run-length, ⟨λ⟩, are also affected by $N_{m}$. We observed an effectively monotonous increase of $\left\langle \tau_{p}\right\rangle$ and ⟨λ⟩ with an increase of $N_{m}$ up to $N_{m}=13$. Following the experimental analysis shown in Figure 4C, we computed, separately, the mean longitudinal velocity of minus-end and plus-end directed steps (Figure 7C) and found a monotonous decrease of the mean minus-end directed velocity with increasing $N_{m}$, effectively saturating above $N_{m}≃7$, similar to the trend seen in the experimental results; the velocity of the plus-end directed steps was not sensitive to $N_{m}$. Noting that Figure 7B,C can be regarded as a “parametric presentation” of $\langle \upsilon_{x}\rangle \;|\;\upsilon_{x}>0$ vs. $\left\langle \tau_{p}\right\rangle$, with $N_{m}$ serving as the parameter, we present in Figure 7D the direct dependence of $\langle \upsilon_{x}\rangle \;|\;\upsilon_{x}>0$ against $\left\langle \tau_{p}\right\rangle$. The similarity to Figure 4E is striking, giving extra support to our interpretation of the experimental results.

To elucidate the connection between these results and the actual number of motors participating in the motion, in Figure 8A, we depict the mean number (per MC unit time)—over all runs—of transient MT-bound motors, $\left\langle M_{B}\right\rangle$, for each$\;N_{m}$ (see Appendix A). As expected, as $N_{m}$ increases, the value of $\left\langle M_{B}\right\rangle$ increases; yet, surprisingly, it effectively saturates at around $\left\langle M_{B}\right\rangle$ =2. We next extracted the fractions of time for different NP “states”; we define an NP state by the number of MT-bound (transporting) motors $M_{B\;}$(Figure 8B and Appendix A). Note that although $\left\langle M_{B}\right\rangle$ ranges roughly between 1 and 2, the distribution of the motor numbers is wide and shows that there is a significant contribution of the one-, two-, and three-motor states, whereas the contribution of $M_{B}$-states of four motors and above is negligible. Figure 8C shows the corresponding mean velocity for each state (see also Appendix A), avoiding NP temporal velocities associated with transitions between these states via binding–unbinding events. Surprisingly, the mean velocity of a particular $M_{B}$-state increases with the increase in $N_{m}$, rather than remaining constant. This can be rationalized by noting that an $M_{B}$-state still corresponds to several microscopic configurational states. At larger $N_{m}$, i.e., smaller NP anchoring distances, most of these microscopic states are associated with MT-bound motors whose NP anchoring distance ξ is smaller. This implies a reduction in polymer tension, i.e., suppression of the elastic coupling between MT-bound motors. Indeed, such an effect has been observed in previous theoretical studies [32,33,36,42] and is associated with the fact that a forward-pulling force on a lagging motor, while somewhat increasing its temporal velocity, has a much smaller effect than the reduction in temporal velocity due to a backward-pulling force on a leading motor. The present analysis confirms that the reduction in the longitudinal velocity with increasing $N_{m}$, which we observed both experimentally and in the simulations, resulted from the increased number of the MT-bound motors participating in the motion, $M_{B}$.

As manifested by the STD shown in Figure 7A and Appendix A and by the corresponding histograms in Appendix A, respectively, the width of the NP longitudinal velocity distribution varies with $N_{m}$. We associate this distribution width both with the width of the velocity distribution at each $M_{B}$-state (not shown here) and with the probability distribution (i.e., time fractions) of these states (Figure 8B), which is sensitive to the value of $N_{m}$. For $N_{m}\geq 2$, we unexpectedly observed a second peak around $\upsilon_{x}=0$, which becomes more pronounced as $N_{m}$ increases (Appendix A). This phenomenon is more strongly demonstrated by the equivalent histograms corresponding to a single MC time-step (Appendix A). We associate this new peak to the growing abundance of states $M_{B}\geq 2$ with increasing $N_{m}$ (Figure 8C). These states are prone to “jamming configurations”, i.e., configurations wherein the “jamming events” (controlled by motor–motor excluded-volume interactions, see Appendix A) dominate.

From this analysis, we conclude that the increase in $N_{m}$ leads to a non-linear increase of$\;\langle M_{B}\rangle$, which, in turn, leads to longer run-times,$\;\langle \tau_{p}\rangle$, and diminishing $\left\langle \upsilon_{x}\right\rangle$. Since the effect on the run-time is more pronounced than on $\left\langle \upsilon_{x}\right\rangle$, the resulting run-length, 〈λ〉, is, in most cases, also enhanced. Moreover, the increase in $\langle M_{B}\rangle$ leads to a slightly narrower NP longitudinal velocity distribution for $N_{m}>4$. While an increase in run-time is expected even without the inclusion of motor–motor coupling [16], the decrease in velocity is a sole consequence of the (elastic and excluded volume) motor–motor interactions. Similarly, the non-trivial change of width of the $\upsilon_{x}$ distribution with increasing $N_{m}$ (STD in Figure 7A) reflects the competition between (i) the increase in $M_{B}$ fluctuations (i.e., the corresponding STD, $\left\langle \delta M_{B\;}\right\rangle$; Appendix A), which acts to increase the velocity fluctuations, (ii) the variability of the velocity fluctuations within each $M_{B}$-state (i.e., $\left\langle \delta v_{x,M_{B}}\right\rangle$; Appendix A), which can be attributed to motor–motor coupling, and (iii) the contributions of the unbinding events, since after each unbinding event an immediate jump of the NP position occurs to balance elastic forces; unbinding events increase with increasing $N_{m}$ (Appendix A and Appendix A).

#### 3.4.2. Angular/Transverse Motion

Consider, next, the angular motion around the MT symmetry axis, which leads to a transverse motion on the projected base plane. As can be seen in Figure 6 and Movies 4–7, the motion is apparently composed of both left- and right-handed helices, combined with large fluctuations. However, on a very long trajectory, the net helical motion (in which the right- and left-handed helices cancel each other) might appear minor and will not reflect the true nature of the motion. Therefore, here, we require very delicate analyses that will reflect both characteristics.

The variation of the mean angular velocity 〈ω〉 and its STD with $N_{m}$ is shown in Figure 9A and in Appendix A, both demonstrating a monotonous decrease with an increase in $N_{m}$. The fact that all values are positive suggests a net right-handed helical motion of the NP for all motor numbers. Combined with a variation of the mean longitudinal velocity with $N_{m}\;$(Figure 7A), and the relationship $\langle \bar{H}\rangle =2\pi \left|\left\langle v_{x}\right\rangle \right|/\left|\langle \omega \rangle \right|$ describing the mean helical pitch size $\left\langle \bar{H}\right\rangle$, this leads to the dependence of $\left\langle \bar{H}\right\rangle$ on $N_{m}$, as shown in Figure 10C, exhibiting an increase in $\left\langle \bar{H}\right\rangle$ for growing $N_{m}$.

As discussed above, this pitch size represents the *net* helical motion, i.e., it includes the cancellations of left- and right-handed helices (as seen in Figure 6 and Movies 4–7). To refine the helical motion analyses, we define a helix (be it left- or right-handed) whenever the angular motion completes a full round (i.e., ϕ=±2π). Note that, even within a single full round, the motion consists of large and frequent right- and left-handed fluctuations.

In Figure 10A, we show the resulting mean pitch size for left- (ϕ=−2π) and right- (ϕ=+2π) handed helices separately, and the mean pitch size regardless of the helix vorticity. Notably, the pitch sizes obtained for these three definitions are all comparable to each other and are much shorter than those deduced using the mean angular velocity (Figure 10C)—all increase with increasing $N_{m}$. To complement these results, we also present, in Appendix A and Appendix A, the mean angular velocity separately for left- and right-handed motion.

#### 3.4.3. Longitudinal and Angular/Transverse Motion Are Correlated

To gain further insight into the complex motion of the NP, we determined whether the longitudinal and angular motions are correlated. In Figure 9B, we dissect the angular velocity for forward and backward steps associated with MC time-step (i.e., positive and negative longitudinal velocities, respectively). Both angular velocity types, associated with either forward or backward steps, show a decrease with increasing motor number $N_{m}$. In addition, the results show that the mean angular velocity is greater (more positive) for backward steps, with the single motor NP showing the highest value. This finding implies that when a backward step is performed, the NP is likely to move to the right. For comparison, we also plot, in Figure 9C, the same distributions on the experimental time interval (0.27 s), showing that these correlations are suppressed, highlighting the importance of the time interval for making the basis for comparison between different results, be it experimental or theoretical.

#### 3.4.4. Binomial Distribution of the Number of NP-Bound Motors, $N_{m}$

For a site-independent binding process, we may assume that the binomial distribution of $N_{m}$ (between the different NPs) holds. Using this distribution, in Appendix A, we computed some of the reported motility characteristics for a varying $\left\langle N_{m}\right\rangle$. For the mean longitudinal and transverse velocities calculated *on the MC time scale*, this analysis shows a small increase in their values (relative to the values for a deterministic motor number, $N_{m}$), especially for small motor numbers (Appendix A); this phenomenon is due to the contribution from NPs with very few motors ($N_{m}=1,\;2,\;3$), whose velocities are always higher (Figure 7A) and, thereby, contribute more to the mean. However, the trends that we deduced for deterministic number of motors (Figure 7 and Figure 9) are not altered. Importantly, when moving to the experimental time scale (0.27 s), the difference is almost non-discernable (Appendix A). Moreover, the run-times and run-lengths (for which the selected time interval is irrelevant) are even less affected by the $N_{m}$ fluctuations, probably due to the contributions from NPs with a high $N_{m}$, for which these values are much higher.

## 4. Discussion

A comparison between experiments and simulations requires the consideration of two issues: (i) knowledge of the relative NP motor coverage $\theta ={\left(\frac{\xi^{*}}{\xi }\right)}^{2}$, which corresponds to the mean number of NP-bound motors via $\langle N_{m}\rangle =\langle N\rangle \times \theta$, where ⟨N⟩ is the mean number of PEG-NLS polymers per NP. We can reasonably assume that regardless of the [NLS], an identical CE batch and [CE] imply identical θ, which allows a qualitative comparison between different systems, as discussed below; (ii) the expected (wide) binomial distribution of $N_{m}$ between different NPs. We have concluded from simulations that, on the experimental time scale (0.27 s), the effect of the binomial distribution is relatively minor. Hence, we no longer emphasize the difference between (simulation) results for deterministic $N_{m}$ and those corresponding to $\left\langle N_{m}\right\rangle$ using the binomial distribution.

The comparison between experiment and theory shows several similarities regarding the dependence of the longitudinal motion on the number of NP-bound motors. Note that, in the experiment, we estimated the number of PEG-NLS, ⟨N⟩, for the different [NLS] used for the motility assays (Table 1 and Appendix A). An increase in ⟨N⟩, when we use the same CE batch and [CE] (i.e., fixed θ), implies a proportional increase in $\left\langle N_{m}\right\rangle$. Both the model simulations and the experiment show a decrease in the mean longitudinal velocity with increasing $N_{m}$ (Figure 4A and Figure 7A and Appendix A). In the model simulations, the mean longitudinal velocity $\left\langle v_{x}\right\rangle$ gradually levels off when $N_{m}$ is increased, rather than continuing to diminish strongly (Figure 7A). This phenomenon occurs mainly due to the “self-regulation” of the mean number of transporting motors, $\left\langle M_{B}\right\rangle$, which effectively saturates at the value of two (Figure 8A). If $\left\langle M_{B}\right\rangle$ would have continued to increase with increasing $\left\langle N_{m}\right\rangle$, $\left\langle v_{x}\right\rangle$ should have continued to decrease strongly due to motor–motor coupling. A very similar effect can be inferred from the experimental results (Figure 4A and Appendix A). A comparison of $\left\langle v_{x}\right\rangle$ between system I ($815\frac{\mathrm{nm}}{s},$
Appendix A) and the model simulation results (833 nm/s in Appendix A) suggests that the mean number of NP-bound motors in system I should be $\langle N_{m}\rangle ≃1$. Thus, we can assume (consistent with the estimated values of ⟨N⟩≃5, 10, 10, 35) that $\langle N_{m}\rangle ≃1,\;2,\;2,\;7$ for systems I, II, III, IV, respectively. First, we consider systems I and II. The drop in the simulation value of $\left\langle v_{x}\right\rangle$ from $N_{m}≃1$ to $N_{m}≃2$ is about 7%, whereas the experiment yields a 6% drop. Similarly, we observe a drop of about 26% in the simulation value of $\left\langle v_{x}\right\rangle$, from $N_{m}≃2$ to $N_{m}≃7$, whereas the experiment yields a 32% drop, which can be regarded as a satisfactory agreement. This semi-quantitative agreement provides indirect evidence that $M_{B}$ is “self-regulated” in the experiment, as it is in the simulations.

As discussed in the Results section, in Figure 4E, we presented the experimentally observed dependence of $\langle \upsilon_{x}\rangle \;|\;\upsilon_{x}>0$ against $\tau_{p}$. We suggested that this behavior could emerge from a parametric dependence on the experimentally hidden variable $\left\langle N_{m}\right\rangle$. Since $\left\langle N_{m}\right\rangle$ is a controlled parameter in the simulations, in Figure 7D, we combined the data presented in Figure 7B,C (for $\langle \upsilon_{x}\rangle \;|\;\upsilon_{x}>0$ and $\left\langle \tau_{p}\right\rangle$ against $\left\langle N_{m}\right\rangle$) to obtain the prediction of the simulations for the direct dependence. The similarity between Figure 4E and Figure 7D is evident, providing further support to our interpretation. In addition, the broad scatter of the single NP data, shown in Figure 4E, can be explained by the expected broad (binomial) distribution of $N_{m}$ discussed above; this is due to the small number of $\left\langle N_{m}\right\rangle$ associated with the nanoscale size of the NPs.

One of the most peculiar features of the angular motion, seen only on the MC timescale of simulations (Figure 9B and Appendix A), was its correlation with the direction of the longitudinal motion (i.e., minus-end or plus-end motion along the MT long-axis). We found that plus-end directed steps dictate (mostly) large right-handed steps. However, for much longer time intervals in the simulation (e.g., the experimental timescale 0.27 s), these correlations disappear (Figure 9C) and, consistently, they do not appear in the experimental results. Moreover, this characteristic also appears as a single-motor property, and it diminishes as more motors become available (increasing $N_{m}$). In fact, our single *mammalian* dynein stepping model, which builds on the raw data of Yildiz and co-workers [38], shows that these correlations appear strongly in the single-motor stepping. (Notably, revisiting the results of our single *yeast* dynein stepping model, we found similar correlations [40]). Thus, further investigation of this peculiar motility feature is required to determine whether it is evident in multi-motor complexes.

Our model simulations did not include two main experimental setup constraints and boundary conditions. First, they permitted the free motion of the NP around the MT surface (similar to the bridge-like setup of Yildiz and co-workers [37]), unlike our experimental setup in which the motion might be impaired due to the presence of the impenetrable glass surface. Second, they do not include MT ends (i.e., MTs are considered infinite) and MT junctions (i.e., only isolated MT tracks are considered), which will be studied elsewhere.

It is important to compare our experimental-theoretical studies to previous results in the literature. Previous gliding assays using kinesin motors [83] found that, as the density of motors increases, the gliding velocity of the MTs decreases, which is in qualitative agreement with our findings. This phenomenon is expected when the load-sharing effect, trending for the opposite, is negligible. However, quantitative comparisons with gilding assays are, in general, not appropriate since the moving MT in a gliding assay is effectively a 1D “cargo” (as studied theoretically in [32]), whose geometry facilitates the fast binding of dynein to the MT if the unbinding of an internal motor occurs; binding and unbinding processes occur mainly at the MT ends—binding at the plus-end and unbinding at the minus-end—as the MT moves. To the best of our knowledge, there is no other systematic experimental study of the effects of motor crowding on spherical cargo velocity; a velocity reduction due to motor crowding was indeed found in previous, albeit more simplified, theoretical studies [16,49,50].

## 5. Conclusions

In this paper, we report a combined experimental and theoretical study of NPs that are carried by multiple *mammalian* dynein motor-proteins. We studied different motility characteristics of these multi-motor NPs and compared them to the known and well-studied case of the single dynein [6,17,18,19,20,38,39,85,89,90,91]. We focused on a single NP size (R=20 nm), although the complexity of motor–motor interactions suggests that motility properties would be strongly affected by varying the NP size; this investigation will take place in future studies.

Although the number of transporting (MT-bound) motors, $M_{B}$, was found to increase with the increasing number of NP-bound motors, $N_{m}$, it remains at a mean value of ≤2 with dominating states $M_{B}=1$, 2, 3. Thereby, the NP can achieve the following motility features: (i) longer run-times and run-lengths than those of a single motor (simulations and experiment); (ii) substantial angular velocity (simulations and experiment), resulting in pronounced helical motion (simulations), yet with a pitch size longer than the single-motor one; (iii) significant longitudinal velocity (simulations and experiment), which is somewhat reduced compared with the single-motor velocity; (iv) plus-end directed motional intervals that are strongly correlated with large right-handed motion (simulations), especially during the temporal state of a single transporting motor; and (v) a maximum width of the longitudinal velocity distribution at an intermediate number of NP-bound motors (simulations and experiment).

Some native cargos are rigid, such as the (cores of) HIV [13], herpes-simplex virus [92], and adenoviruses [14,15], and super-coiled plasmids [93]; others are relatively flexible, as in the case of cytoplasmic lipid granules (i.e., native liposomes). Our current model simulations, as well as previous theoretical results [32,33,42] and the present motility assays of a rigid NP, demonstrate that, for rigid cargos, a maximal transport efficiency can be achieved when flexible linkers mediate between the rigid NP body and the motor proteins, allowing the NP to optimally use the viable motors. We, thus, conjecture that rigid native cargos use a similar mechanism for their active intracellular transport. For instance, it is possible that the abundance of disordered loops in the NFkB structure gives it extra flexibility for the enhancement of the super-coiled plasmid motility [93].

If used intracellularly, the ability of the NP to make long trajectories enhances its active transport towards the nucleus, which is relevant both to the rigid core viruses mentioned above and, possibly, to drug delivery applications using nano cargos [16,94,95]. In addition, by maintaining its helical motion—and, in particular, by making large right-handed steps when stepping backward—we posit that the NP is capable of bypassing obstacles on the MT tracks or in their vicinity. Indeed, this suggestion (albeit without mentioning the backward-right-handed correlated motion) has already been put forward [37] and was further observed in (surface-free) bead motility assays [96]. We plan to further investigate these hypotheses in future work.

## Figures and Tables

**Figure 1 ijms-22-08893-f001:**
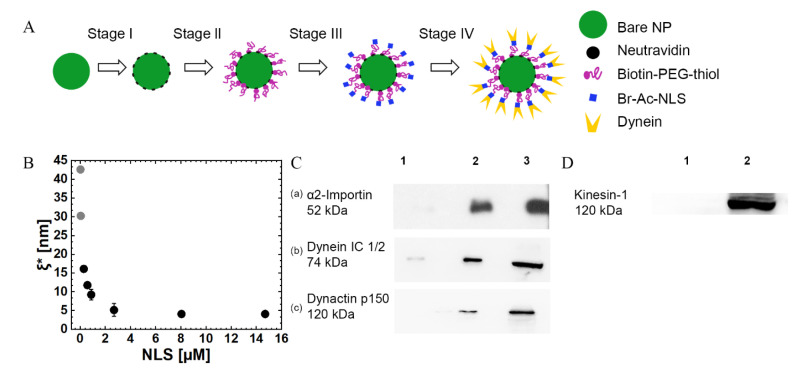
NP synthesis and characterization. (**A**) The NP synthesis process entails several consecutive steps, where, at each step, a single component is added. (**B**) Mean anchoring distance between neighboring PEG-NLSs $\xi^{*}$, against the concentration of TAMRA-NLS (shown as NLS (*x*-axis) in the figure). The grey dots correspond to extrapolated values that were calculated from the fit of <*N*> vs. <NLS> (see Table 1, Appendix A). Error bars indicate the standard deviations for three experiments. (**C**) Western blot (WB) analysis results, demonstrating the recruitment of α2-importin (a); *mammalian* dynein motors (b); and dynactin to the NPs after incubation in a Hela cells extract. Group 1 refers to NPs coated with Biotin-PEG-thiol, group 2 refers to PEG-NLS coated NPs, and group 3 refers to Hela cell extracts without NPs. (**D**) WB analysis results, showing that kinesin-1 does not bind the PEG-NLS coated NP. Group 1 refers to PEG-NLS coated NPs, and group 2 refers to Hela cell extracts without NPs.

**Figure 2 ijms-22-08893-f002:**
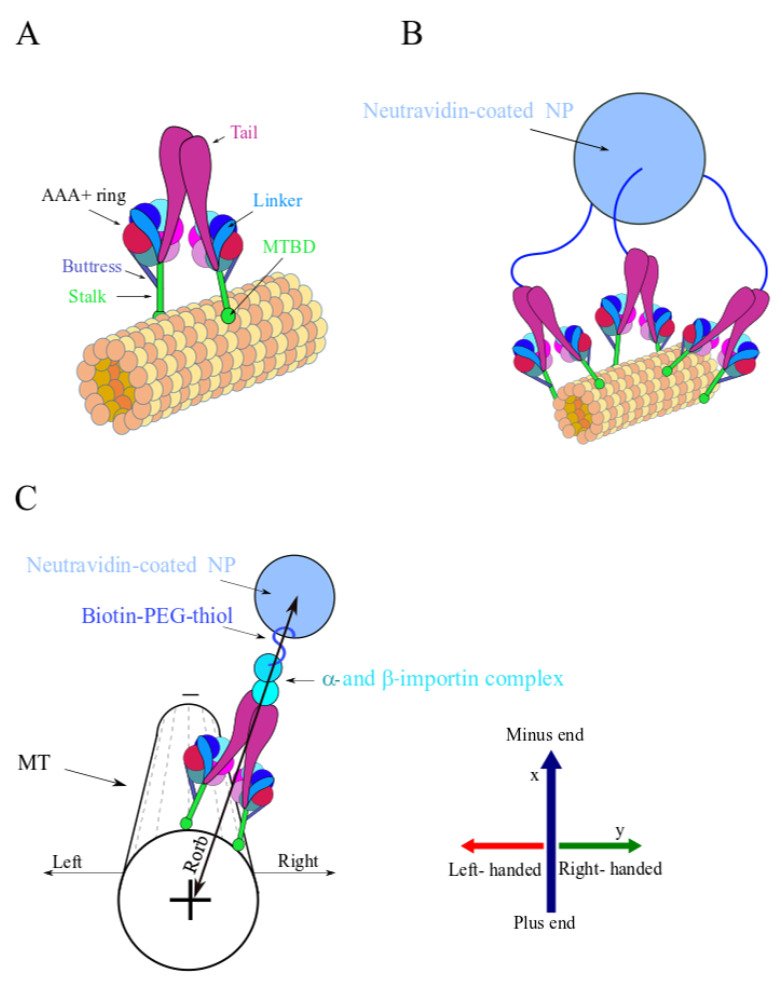
Illustration of single dynein and multi-motor NP (not to scale). Some biological factors, such as dynactin and adaptors, are not shown although they are necessary for the activity of the motor. (**A**) Single dynein motor-protein bound to the MT surface. The MT protofilaments and the corresponding α and β tubulin subunits are represented by the collection of orange and yellow spheres. Color coding: linker (light blue), tails (magenta), the stalks and microtubule binding domains (MTBD) (green), buttress (blue), and the AAA+ ring (red). (**B**) A simplified illustration of an NP with three anchored PEG polymers over its surface, each of which is connected to a single dynein. The Neutravidin-coated NP is represented by a light-blue sphere and the PEG polymers are represented by the blue curved lines. (**C**) Illustration of the spatial orientations used throughout the article to characterize NP motion, i.e., longitudinal motion towards the MT plus- or minus-end and left or right transverse motion with respect to the MT long axis. The Neutravidin-coated NP color code is as in (**B**); the α- and β -importin complex is also shown. The MT is presented with fewer details, where the dashed lines represent the MT protofilaments. Note that the longitudinal axis is denoted by *x* throughout the text and the transverse axis is denoted by *y*; positive and negative *x* correspond to MT minus- and plus-end directions, respectively. Positive *y* is the right direction and negative *y* is the left direction.

**Figure 3 ijms-22-08893-f003:**
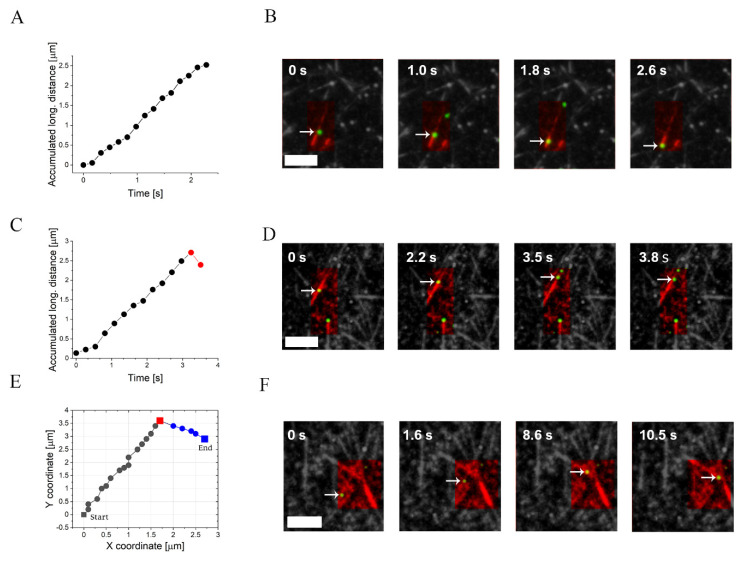
Examples of the NP motion modes. The two first rows show the accumulated longitudinal distance of an NP (marked by an arrow) and the corresponding snapshots for different motion modes. (**A**,**B**) Directed motion of an NP moving towards MT minus-end (marked-end MTs are shown). (**C**,**D**) Minus-end directed motion with a backward (plus-end) step at the end of the NP trajectory. (**E**,**F**) Trajectory of an NP (marked by an arrow) traversing between crossing MT tracks. The MT tracks are colored in red and the NPs in green. Conditions: [NLS]=0.05 μM, [CE] = 3.4 mg/mL (system III). The interval between frames is 0.27 s. The mean radius of the bare NP is 20 nm. Scale bars are 5 μm in all images.

**Figure 4 ijms-22-08893-f004:**
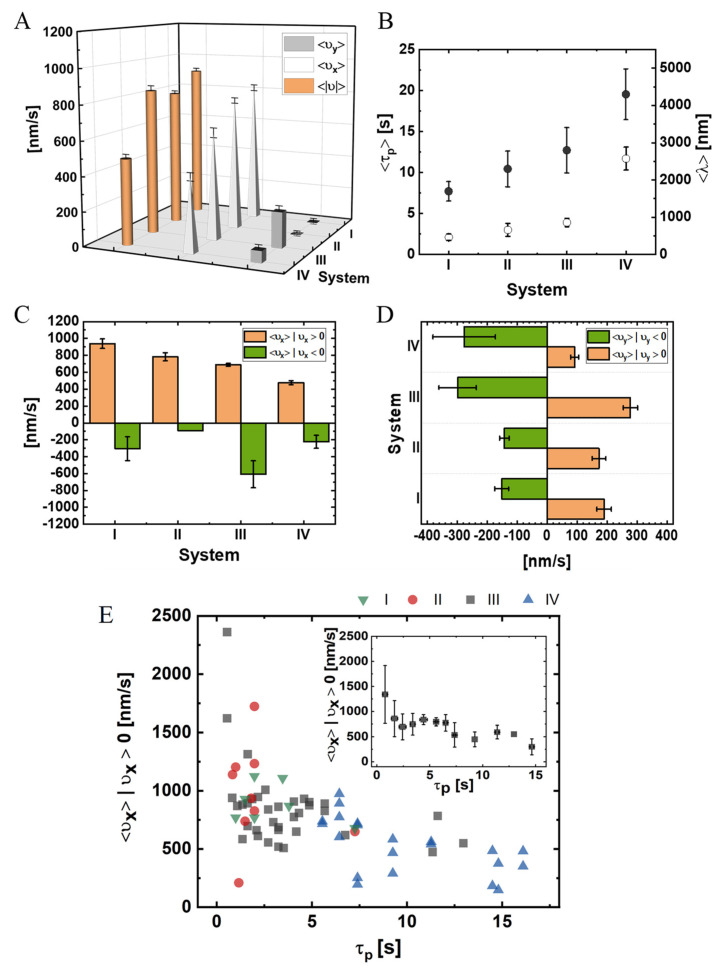
NP experimental mean velocities, run-times, and run-lengths for systems I–IV (data for A–D are provided in Appendix A). (**A**) Mean longitudinal, $\left\langle \upsilon_{x}\right\rangle$, transverse, $\left\langle \upsilon_{y}\right\rangle$, and absolute, $\langle \upsilon \rangle =\sqrt{\left(\upsilon_{x}^{2}+\upsilon_{y}^{2}\right)}$ velocities. A minus-end directed motion corresponds to $\upsilon_{x}>0$ and a right-handed motion corresponds to $\upsilon_{y}>0$ (see Figure 2C for spatial orientation). (**B**) Run-times and run-lengths: the run-time is measured from the time the NP binds to the MT until the time it unbinds from it; the run-length is the accumulated distance traveled along the MT symmetry axis; $\upsilon_{x}>0$ and $\upsilon_{x}<0$ define minus-end and plus-end directed motion, respectively. (**C**) Mean longitudinal velocity, evaluated separately for minus-end directed and plus-end directed motions. (**D**) Mean transverse velocity, evaluated separately for right-handed and left-handed motions. (**E**) Main panel: $\langle \upsilon_{x}\rangle \;|\;\upsilon_{x}>0$ is plotted against $\tau_{p}$. Each data point corresponds to a specific NP, colors correspond to the experimental system (I–IV). Inset: average values over all NPs, regardless of their associated system. Values indicate mean±SEM. The mean radius of the bare NP is 20 nm.

**Figure 5 ijms-22-08893-f005:**
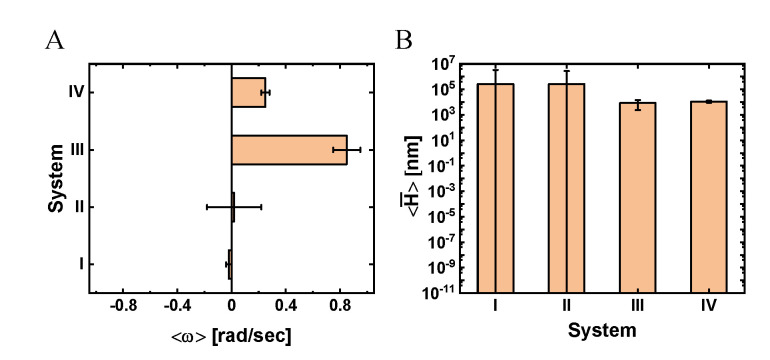
Mean angular velocity and helical pitch size for systems I–V. (**A**) Mean angular velocity, ⟨ω⟩. (**B**) Mean helical pitch size, $\left\langle \bar{H}\rangle =\frac{2\pi }{\left|\langle \omega \rangle \right|}|\langle \upsilon_{x}\right\rangle$. Values indicate mean±SEM. The mean radius of the bare NP is 20 nm. For further details, see Materials and Methods and Appendix A.

**Figure 6 ijms-22-08893-f006:**
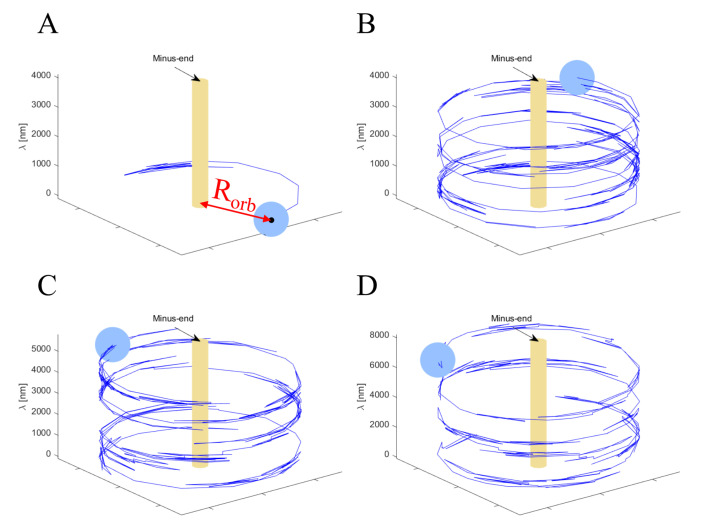
Traces of selected simulated trajectories. The light-blue sphere represents the NP (R=20 nm), the blue trailing lines represent the trajectory, and the yellow cylinder represents the MT. Note that while the MT and NP radii are drawn to scale, the MT length is not. (**A**) Initial trace of a single motor trajectory. The red arrow marks the distance between the MT and NP centers, $R_{\mathrm{orb}}$. (**B**–**D**) Traces of selected trajectories of an NP with $N_{m}=1,\;3,\;\mathrm{and}\;13$ motors, respectively.

**Figure 7 ijms-22-08893-f007:**
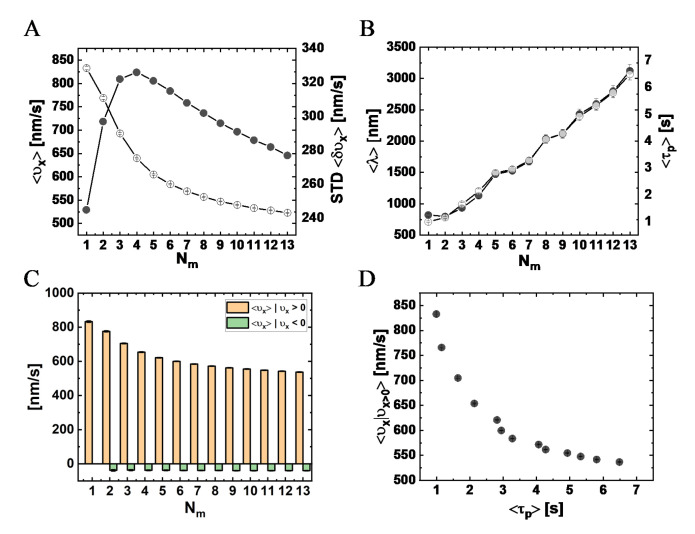
Simulation results for the mean longitudinal velocity, $\left\langle \upsilon_{x}\right\rangle$, run-length, ⟨λ⟩, and run-time, $\left\langle \tau_{p}\right\rangle$ for different ($R=20\;\mathrm{nm},\;N_{m})$ configurations, for a time interval Δt=0.27 s. (**A**) The data for the mean (±SEM ) longitudinal velocity (black dots) and STD (empty circles) are provided in Appendix A see Appendix A and Appendix A for a Δt= MC step time. (**B**) Mean (±SEM ) run-length (black dots) and run-time (empty circles), using the data provided in Appendix A. (**C**) Mean (±SEM ) instantaneous longitudinal velocity, $\left\langle \upsilon_{x}\right\rangle$, separated for minus-end directed, $\upsilon_{x}>0$, and plus-end directed, $\upsilon_{x}<0$, directions; data are provided in Appendix A. (**D**) Mean (±SEM ) instantaneous minus-end directed longitudinal velocity $\langle \upsilon_{x}\rangle \;|\;\upsilon_{x}>0$ against mean run-time $\left\langle \tau_{p}\right\rangle$.

**Figure 8 ijms-22-08893-f008:**
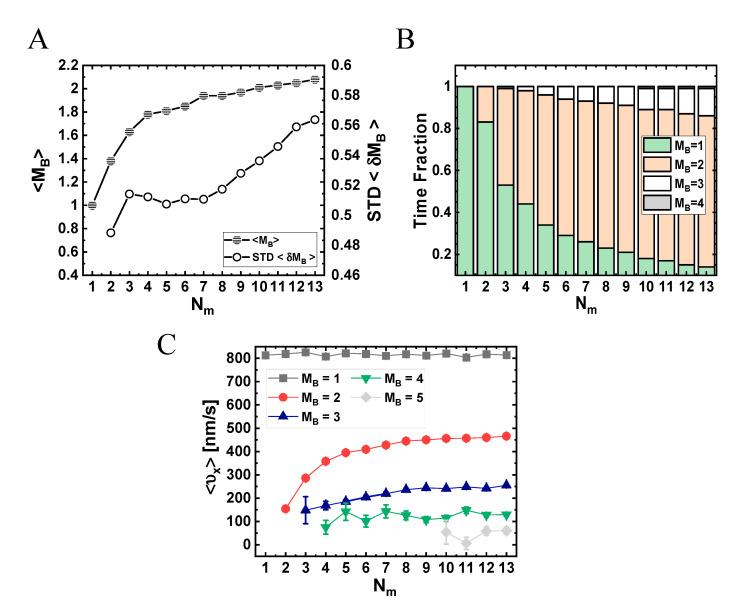
Participating (MT-bound) motors, $M_{B}$, for different ($R=20\;\mathrm{nm},N_{m})$ configurations and a time interval Δt = MC-step time. (**A**) Mean (±SEM) number of participating motors $\langle M_{B}\rangle$ and STD against the number of NP-bound motors$\;N_{m}$ (data are shown in Appendix A). (**B**) Fraction of time the NPs spend in each of the $M_{B}$ -states, for different values of $N_{m}$ (data are shown in Appendix A). (**C**) Mean (±SEM ) longitudinal velocities of the different $M_{B}$ -states for different values of $N_{m}$ (data are shown in Appendix A). Note that states of $M_{B}\geq 5$ are extremely rare for $N_{m}\leq 13$; therefore, in (**B**), the state $M_{B}=5$ is not shown and it appears only for $N_{m}\geq 10$ in (**C**).

**Figure 9 ijms-22-08893-f009:**
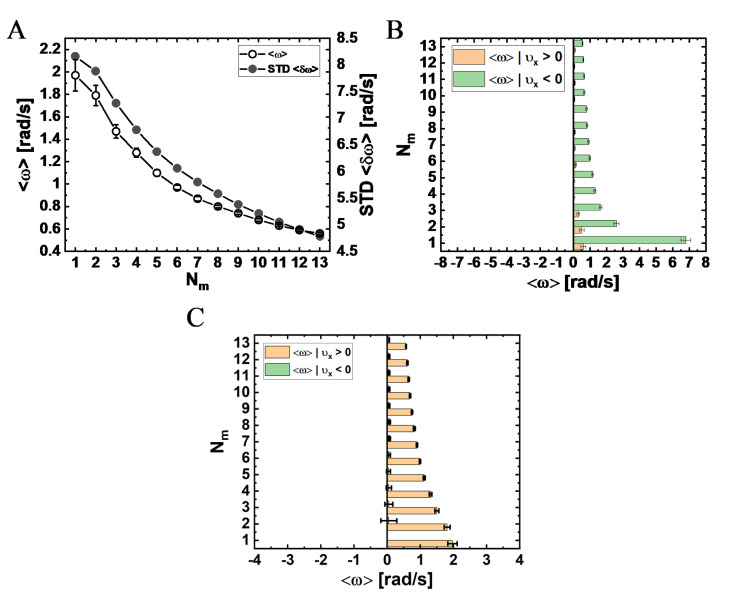
Mean angular velocity, ⟨ω⟩, as a function of NP-bound motors, $N_{m}$, for NPs with a radius of R=20 nm. (**A**) Mean angular velocity, ⟨ω⟩, and the corresponding STD, against $N_{m}$, for time interval Δt=0.27 s  (data are provided in Appendix A; see Appendix A and Appendix A for a Δt= MC step time). (**B**,**C**) Mean (±SEM) angular velocity calculated separately for minus-end directed motion, $\langle \omega \rangle \;|\;\upsilon_{x}\rangle 0$, (orange bars) and plus-end directed motion, $\langle \omega \rangle \;|\;\upsilon_{x}<0,$ (green bars), against the number of NP-bound motors, $N_{m}$, for a time interval Δt that equals to the MC-step time (**B**); data are provided in Appendix A and 0.27 s (**C**); data are provided in Appendix A).

**Figure 10 ijms-22-08893-f010:**
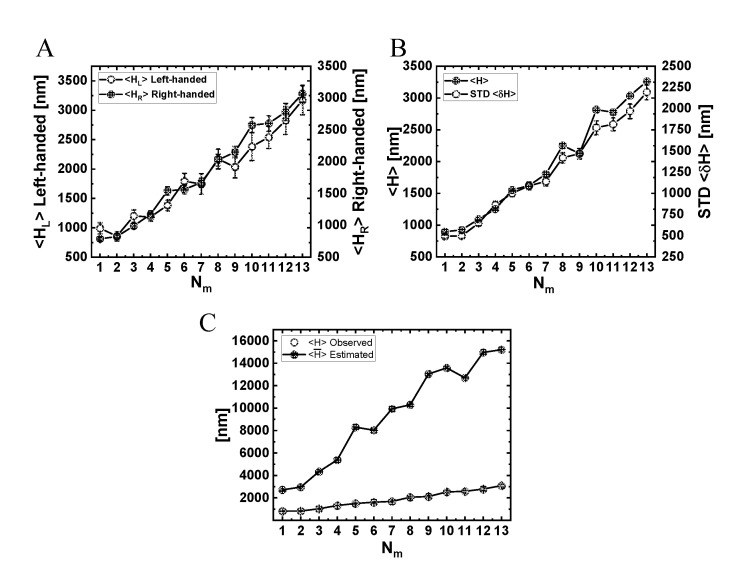
Analysis of the actual NP helical motion from observed trajectories. (**A**) Mean (±SEM) and STD helical pitch size, 〈H〉, evaluated separately for left- and right-handed helical vorticities (data are provided in Appendix A). (**B**) Mean and STD of the helical pitch size, regardless of the helical vorticity (data are provided in Appendix A). (**C**) Comparison between the actual helical pitch size (from the observed simulated trajectories) and the helical pitch size estimated using the mean values of angular and longitudinal velocities, $\langle \bar{H}\rangle =\frac{2\pi }{\left|\langle \omega \rangle \right|}\left|\left\langle \upsilon_{x}\right\rangle \right|$ (data are provided in Appendix A).

**Table 1 ijms-22-08893-t001:** Summary of the systems studied and their basic properties: NLS and CE concentrations, estimated mean number ⟨N⟩ of PEG-NLS per NP, estimated mean anchoring distance $\xi^{*}$, and fractions of NPs performing distinct modes of motion. An identical cell extract was used in systems I and II, and another extract was used in systems III and IV. The bare NP mean radius is 20 nm.

**System**	**I**	**II**	**III**	**IV**
NLS conc. [µM]	0.025	0.05	0.05	0.3
CE conc. [mg/mL]	3.4	3.4	3.4	3.4
⟨N⟩	5.2	10.4	10.4	36.6
$\xi^{*}$ [nm]	42.6	30.2	30.2	16.1
**Motion mode**	**Fraction**			
	**I**	**II**	**III**	**IV**
Minus-end directed motion; detachment before arrival to the end.	0	0	0.64	0.72
Minus-end directed motion; NP stuck when MT-end is reached.	0.11	0.24	0.08	0.20
Minus-end directed motion + a single backward (plus-end) directed step, either during the motion or when the MT-end is reached.	0.87	0.52	0.23	0
MT track traversing	0.02	0.24	0.05	0.08

## Data Availability

The data presented in this study are available on request from the corresponding authors.

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
