# Peer review of "Nano-Particles Carried by Multiple Dynein Motors Self-Regulate Their Number of Actively Participating Motors"

_ijms, 2021, doi:10.3390/ijms22168893_

Round 1

Reviewer 1 Report

Halbi et al. have presented an extensive study on self-regulated motility of nanoparticles labeled with dynein motors on microtubule surfaces.  They have analyzed the effects of the number of dynein motors on the velocity, microtubule bound motors, run time, run length, and several other factors both experimentally and using simulations.  It is evident that the authors have performed a thorough and extensive analysis of the system to highlight the different aspects affecting 2D motility of dynein on a microtubule carrying a nanoparticle (cargo). 

The topic is interesting and, for the most part, the manuscript is well-written and organized.  The addition of the movies as supplementary material is very helpful in visualizing the data presented and discussed.  That being said there are a few areas that I would like clarification and some general interest questions:

The authors mention that the average number of motors bound to the microtubule surface is 2.  I am curious to know if this is the average number bound at one given time or if it is the number of motors that have contact with the microtubule at any time during their "walk".  My question stems from the thought that, since there are several dynein motors available to bind to the MT from a single NP, if the motors are able to bind/unbind (full dissociation of MTBDs and reassociation of another or the same motor) then is it possible that the mobility is influenced in some part by a wheeling effect by different motors repeatedly binding/unbinding from the NP? The authors mention that contributions in the unbinding events results in an immediate jump of the NP position and that these unbinding events increase with the increasing Nm. 

Additionally, the movies provided seem to show that there is only one NP traversing a given MT. Did the authors ever look at or analyze any cases where more than one NP cargo was being carried on the same MT?

Have the authors looked into different lengths of the PEG and how this may affect the mobility? How was the size of PEG decided? On line 186, the authors mention that the Biotin-PEG-thiol molecules are in the "mushroom regime".  I am not familiar with the term nor the significance of this statement.  If the authors could please clarify?

Other suggestions:

Please add specification of the type of lasers used for the TIRF analysis.

Line 117:  What do the authors mean by "they"? Originially, after reading the first sentence I thought the authors were defining "they" to be their NP but with the third sentence I am believe "they" may refer to other studies.  Please define "they" to avoid the confusion.

There are a few cases in the results section where the verb tense changes.  For example, line 171 written in the present tense while similar surrounding statements are written in the past tense.

Line 258:  The authors mention "twice larger concentration" but it is not clear which concentration is twice larger (CE or NLS).  From the SI table it is clear that they are referring to the [CE], however, it would be helpful to clarify this in the main text as well.

Similarly, the sentence on line 294 is a bit confusing. Perhaps editing it to say "…tested the influence of NLS conc. on NP motion within the same CE batch perhaps?.."

Overall, I suggest a read through of the manuscript as a whole to correct some minor article omission or wording.  For example, line 321 should read "increases with" instead of "with increasing" and line 325 should read "due to the" instead of "due the".

Acronyms such as SEM and STD have not been defined in text.  I assume these refer to standard error measurement and standard deviation.

In the SI, the sentences presented before SI Eq. 1.1 talk about adsorption of TAMRA-NLS molecules that did not absorb to the NPs but I believe this sentence should read "absorption of the remaining TAMRA-NLS molecules that did not adsorb to the NPs"

Author Response

Thank you very much for your valuable comments. Please check our responses in the attachment.

Reviewer 2 Report

The current study focuses on elucidating how motor-motor interactions (with the example of dynein) influence the motility on the single microtubule level. Using experimental approaches (with bead-motility assays) and model simulations, the authors highlighted left- and right- handed helical motion of nano-particles (NP), and self-regulation of the number of microtubule-bound motors, actively transporting. As reviewer, I agree to say that their manuscript is very interesting, in particular because all of their results (both experiments and simulations) could support for the first time helical motion of motors such as dynein on microtubules. However, despite this as well as the quality of the English and writing in the manuscript, many points concerning both the clarity of the manuscript and important details (experimental or scientific) detract from the quality of this study. Thus, all of these preclude acceptance the article for publication in Cancers.

1/ in the section Abstract: overall, the summary is well written and sets out the major facts of the study. I am more critical about the last sentence (p1 - lines 33-34: "that can enhance active transport efficiency when facing the crowded cellular environment") in which the authors go too far on the hypothesis of a better efficiency of motor motion on MTs if this is helical, in a crowded environment such as the cytoplasm of cells. To be sure, it would have been necessary to reproduce this bulk (such as with the addition of kinesins for example).

2/ in the section Introduction: I find it a bit long; and to start by presenting figures (p3 - lines 101-110, Fig 1A and 2A-B) in this section is not very conventional, although I understand the authors who wish to present their study model: thus, I suggests that this whole intro part be placed as a preamble to the Result section. Minor comments:

  • p4 lines 155-163: what is the use of presenting this data in the intro section? Please, specify;
  • Please, define the term 'NP' in the intro section (here, only presented in the abstract)

3/ in the section Materials and Methods:

  • Part §4.4.4. NP imaging using Total internal reflection fluorescence microscope (TIRFM): please, complete the description with the types of emission filters used. Moreover, is it a system that kept the MTs at 37 ° C (this information is not clearly indicated in §4.4.2 and 4.4.3). If yes, specify it somewhere.

4/ in the section Results:

  • P4 line 181: if we look closely at fig 1B, it is even a saturation greater than 3 µM for PEG-NLS
  • P4 line 182: is it possible to indicate or to calculate the gyration radius for the PEG-NLS model? or is it very close to the Biotin-PEG-thiol model already presented? if so, this would allow the reader to appreciate the difference between the 2 models (beyond an NLS sequence for the directionality of their NP);
  • I propose to move fig 2 (p6) as supplemental figure, I agree this figure is important and it can be commented on by the authors, but it is not a result as such; moreover, please better explain what the reader sees (color code) on schema of a NP-motor complex in fig 2A;
  • Table 1 (and M&M section too): it seems to me very unconventional, even hazardous, to calculate a protein concentration (even by the Bradford method), then to use it in this approach, whereas we will find in this extract thousands of proteins, including tubulin (not to mention all the problems of extraction, sequestration of proteins in membranes, etc.). Is it not possible to extract only the motor proteins (if not, use a recombinant form of the proteins)? As the authors write, the protein composition from batch to batch can vary wildly, although the protein concentration ('CE' in experiments) does not appear to vary. Suddenly, this undermines the reproducibility of their results: it would be desirable to increase the number of tests in order to have a more robust mean value (+ SD) for the comparison between the experimental conditions.
  • P7 lines 262-263: concerning the movies 1 - 3, the frame rate is too fast so that we can see that the authors have added a small arrow pointing a NP on a raw: please, add this white arrow all the time for monitoring the NPs;
  • P7 Fig 3 + Table 1: it would be desirable for the authors to supplement their results with a statistical analysis of the 3 modes of motion: it would be easier to see if all the modes are equivalent in frequency; on the images, please add an arrow pointing to the NP that the authors follow (in particular for the fig 3 E-F);
  • P9 l 323-324: the authors hypothesize that the decrease in NP velocities comes from inter-motor interactions, to explain the differences between systems I-II and III-IV. It is more likely that this is due to spatial hindrance of numerous NP on MT, or even a saturation of binding sites on the TMs, right? To discuss maybe;
  • Fig 4: The whole part dealing with longitudinal motion is relevant and complete. On the other hand, the study on angular / transverse motion is subject to discussion. Indeed, the lateral resolution of TIRF microscopy (even with sub-pixel resolution and 10 nm scale precision of the 'Particle tracking' computer module) is insufficient: the apparent width of an isolated MT (which is not guaranteed with MT reconstituted as in the study) is between 200 nm and 300 nm (dimensions obtained with a super-resolved system such as STORM or PALM, and not with TIRF!), this is mainly due to the irradiance of the fluorophores labeling MTs. So, how do the authors explain transverse motions > 800 nm as for system IV? Please, discuss this point;
  • The model and the simulation of motions (longitudinal, angular, and transverse) are very interesting. They provide a better understanding of the experimental data. These parts are relatively long all the same and not always easy for the novice to follow the description of the figures, especially since there are many points discussed in these descriptions of results. This could be improved by the authors (by transferring these parts discussed in the Discussion section, for example), as one would expect the more the number of NP increases on the MT the more the motion is difficult, probably due to this spatial hindrance on the MT.

5/ in the section Discussion: the authors recapitulate their main results and comment on them (again for some) without putting them in perspective with the knowledge on this subject. Is it finally in agreement with the data in the literature? Could we compare their data with other motors moving on MT? On other cytoskeletons?

6/ in the section Conclusions: there are a lot of items here that should be placed in the discussion section

7/ Supplementary Informations (Figures, Tables, References): no comment, all elements are relevant.

Author Response

(The authors gave the same response as above.)

Round 2

Reviewer 2 Report

The current study focuses on elucidating how motor-motor interactions (with the example of dynein) influence the motility on the single microtubule level. Using experimental approaches (with bead-motility assays) and model simulations, the authors highlighted left- and right- handed helical motion of nano-particles (NP), and self-regulation of the number of microtubule-bound motors, actively transporting. As I already said in my reviewing report, your work are very interesting, in particular because of your experiments and simulations results supporting for the first time helical motion of dynein motors on microtubules. Also, you have answered all of my questions and made the necessary changes to the manuscript. Therefore, I give a very favorable opinion for the publication of your article. 
